# Clinical and Pathologic Features of Congenital Myasthenic Syndromes Caused by 35 Genes—A Comprehensive Review

**DOI:** 10.3390/ijms24043730

**Published:** 2023-02-13

**Authors:** Kinji Ohno, Bisei Ohkawara, Xin-Ming Shen, Duygu Selcen, Andrew G. Engel

**Affiliations:** 1Division of Neurogenetics, Center for Neurological Diseases and Cancer, Nagoya University Graduate School of Medicine, Nagoya 466-8550, Japan; 2Department of Neurology and Neuromuscular Research Laboratory, Mayo Clinic, Rochester, MN 55905, USA

**Keywords:** congenital myasthenic syndromes, neuromuscular junction, muscle nicotinic acetylcholine receptor, cholinesterase inhibitors, ephedrine, salbutamol (albuterol), amifampridine

## Abstract

Congenital myasthenic syndromes (CMS) are a heterogeneous group of disorders characterized by impaired neuromuscular signal transmission due to germline pathogenic variants in genes expressed at the neuromuscular junction (NMJ). A total of 35 genes have been reported in CMS (*AGRN, ALG14, ALG2, CHAT, CHD8, CHRNA1, CHRNB1, CHRND, CHRNE, CHRNG, COL13A1, COLQ, DOK7, DPAGT1, GFPT1, GMPPB, LAMA5, LAMB2, LRP4, MUSK, MYO9A, PLEC, PREPL, PURA, RAPSN, RPH3A, SCN4A, SLC18A3, SLC25A1, SLC5A7, SNAP25, SYT2, TOR1AIP1, UNC13A, VAMP1*). The 35 genes can be classified into 14 groups according to the pathomechanical, clinical, and therapeutic features of CMS patients. Measurement of compound muscle action potentials elicited by repetitive nerve stimulation is required to diagnose CMS. Clinical and electrophysiological features are not sufficient to identify a defective molecule, and genetic studies are always required for accurate diagnosis. From a pharmacological point of view, cholinesterase inhibitors are effective in most groups of CMS, but are contraindicated in some groups of CMS. Similarly, ephedrine, salbutamol (albuterol), amifampridine are effective in most but not all groups of CMS. This review extensively covers pathomechanical and clinical features of CMS by citing 442 relevant articles.

## 1. Overview of Congenital Myasthenic Syndromes (CMS)

CMS are caused by defects in molecules expressed at the neuromuscular junction (NMJ) and are characterized by defective neuromuscular signal transduction [1,2,3]. As of January 2023, germline pathogenic variants in 35 genes have been reported (*AGRN, ALG14, ALG2, CHAT, CHD8, CHRNA1, CHRNB1, CHRND, CHRNE, CHRNG, COL13A1, COLQ, DOK7, DPAGT1, GFPT1, GMPPB, LAMA5, LAMB2, LRP4, MUSK, MYO9A, PLEC, PREPL, PURA, RAPSN, RPH3A, SCN4A, SLC18A3, SLC25A1, SLC5A7, SNAP25, SYT2, TOR1AIP1, UNC13A,* and *VAMP1*) (Figure 1). The causative genes can be classified into 14 groups depending on the pathomechanical features. Clinical features and therapeutic strategies are shared between some or all groups of CMS, but some features and therapies are unique to specific groups of CMS. Some therapies are ineffective or even contraindicated in some groups. Clinical features and therapeutic responses are generally difficult to predict a defective molecule.

Clinical features of CMS are characterized by muscle fatigue, muscle weakness, muscle hypoplasia, and minor facial anomalies like low-set ears and high-arched palate in some patients. Autoimmune myasthenia gravis (MG) also compromises the NMJ signal transduction but is caused by autoantibodies against the acetylcholine receptor (AChR), muscle-specific receptor tyrosine kinase (MuSK), low-density lipoprotein receptor-related protein 4 (LRP4), or others. In contrast to MG, diurnal fluctuation of muscle strength and muscle fatigue are not always observed in CMS. In some CMS patients, day-to-day fluctuation of muscle strength is prominent. Diurnal fluctuation of external ophthalmoplegia is associated with diplopia in MG, but not always in CMS. This is likely due to the presence of external ophthalmoplegia since infancy, which enables compensation for the fluctuating visual axes. Some lack eye symptoms and are referred to as limb-girdle CMS. Most CMS patients develop the disease before age 2 years, but, in some patients, symptoms develop immediately after birth but temporarily subside thereafter until adolescence or adulthood. Including these neonatal transient patients, CMS can develop at any age including adolescence and adulthood. Some patients with *CHAT*-CMS, *LAMB2*-CMS, *SLC5A7*-CMS, *SNAP25*-CMS, *UNC13A*-CMS, *DPAGT1*-CMS, *ALG2*-CMS, *MYO9A*-CMS, *SLC25A1*-CMS, and *PURA*-CMS exhibit developmental delay. This can be caused by defective cholinergic neurotransmission in the central nervous system (CNS) or hypoxic brain injury due to repeated apneustic attacks, but the exact mechanisms remain to be elucidated. Episodic apnea is frequently reported in *CHAT*-CMS, *COLQ*-CMS, and *SCN4A*-CMS, but is also observed in other groups of CMS. Siblings of CMS patients with episodic apnea sometimes die with a diagnosis of sudden infantile death syndrome (SIDS) [4,5]. Continuous monitoring of apnea is required for these patients.

Most CMS patients show autosomal recessive inheritance or require biallelic pathogenic variants. An autosomal dominant inheritance or a de novo hemiallelic pathogenic variant is observed in slow-channel CMS (SCCMS), *SNAP25*-CMS, *PURA*-CMS, and 4 out of 11 patients with *SYT2*-CMS. *SYT2*-CMS, *SNAP25*-CMS, *VAMP1-*CMS, *UNC13A*-CMS, *RPH3A*-CMS, and *LAMA5*-CMS are characterized by defects in the SNARE complex, and phenotypically similar to Lambert-Eaton myasthenic syndrome (LEMS). Both LEMS and LEMS-like CMS show an increment of compound muscle action potentials (CMAP) in response to high-frequency repetitive nerve stimulation (RNS) or spontaneous muscle contractions. In addition, in five patients in three pedigrees with *AGRN*-CMS, which primarily shows endplate AChR deficiency, a marked increment of CMAP after exercise was reported [6], but not in other *AGRN*-CMS patients. *GMPPB*-CMS, *GFPT1*-CMS, and SCCMS show elevated serum creatine kinase (CK) levels up to 24 times the upper limit of the normal range [7,8].

## 2. Electrophysiology, Muscle Biopsy, Laboratory Examinations, Differential Diagnosis, Epidemiology, Inheritance, and Therapeutic Perspectives

### 2.1. Electrophysiological Examinations

RNS or single-fiber electromyography (SFEMG) is required to diagnose CMS. However, next-generation sequencing technologies have enabled extensive genetic analysis, and a plethora of CMS patients have been diagnosed and reported in the absence of RNS or SFEMG. RNS at 2–3 Hz shows 10% or more decrements of the compound muscle action potential (CMAP).

A single nerve stimulus elicits a repetitive CMAP (R-CMAP) in some patients with SCCMS, *COLQ*-CMS, and *PURA*-CMS. R-CMAP rapidly disappears by RNS or by spontaneous exercise, and a single nerve stimulus after a prolonged rest is required.

In *SCN4A*-CMS, a decrement of CMAP is not induced by low-frequency RNS but by high-frequency RNS. In CMS caused by defective recycling of acetylcholine (ACh) (*CHAT*-CMS, *SLC18A3*-CMS, *SLC5A7*-CMS, and *PREPL*-CMS), decremental CMAP by low-frequency RNS is elicited at rest in some patients, but only after exercise or high-frequency RNS in the other patients.

In LEMS-like CMS caused by *SYT2*-CMS [9], *VAMP1*-CMS [10], *UNC13A*-CMS [11], *RPH3A*-CMS [12], and *LAMA5*-CMS [13], low-frequency RNS causes a decremental CMAP, whereas high-frequency RNS elicits an incremental CMAP [14]. In another form of LEMS-like CMS of *SNAP25*-CMS, low-frequency RNS caused a decremental CMAP, but high-frequency RNS was not examined [15].

In performing RNS, it is essential to fix the recording electrodes well. Muscle twitch by the first electrical stimulus in RNS moves the recording electrodes and decreases the height and area of CMAP, which could be misdiagnosed as a decremental response. The movement of the recording electrodes can be easily detected by a change in the shape of CMAP from the second stimulus.

Single fiber electromyography (SFEMG) has a higher sensitivity than RNS to detect defective signal transmission at the NMJ but has a lower specificity. Although SFEMG is technically challenging, some neurophysiologists diagnosed a large number of CMS patients only by SFEMG [16].

### 2.2. Muscle Biopsy and Creatine Kinase (CK)

Muscle biopsy shows tubular aggregates or rimmed vacuoles in glycosylation defects in *GFPT1*-CMS [17,18,19,20], *DPAGT1*-CMS [21,22,23], *ALG2*-CMS [24], but not in *ALG14*-CMS [24] or *GMPPB*-CMS [25]. These muscle pathologies, however, are not always observed. In a case with *ALG2*-CMS, muscle biopsy at age 14 years showed no tubular aggregates [24]. On the other hand, endplate myopathy in SCCMS was reported as tubular aggregates at the light microscopy level [26] or inclusion body myositis (IBM)-type inclusions at the electron microscopy level [27]. Defects in *GMPPB* cause muscular dystrophy-dystroglycanopathy (MDDG) type 14 [28]. *GMPPB*-CMS similarly exhibits hypoglycosylation of α-dystroglycan, as well as muscular dystrophy [25]. *GMPPB*-CMS also exhibits fibrosis and adiposis of skeletal muscle by MRI [25], as well as centronuclear myopathy [29].

Serum CK levels are normal in most groups of CMS. However, serum CK levels are elevated ~1.5 times the upper limit of normal in endplate myopathies in SCCMS, ~3 times in tubular aggregates in *GFPT1*-CMS and *DOK7*-CMS, and 2-to-24 times (average 10.7 times) in *GMPPB*-CMS [7,8].

### 2.3. Differential Diagnosis

Abnormal muscle fatigue should be differentiated from MG and LEMS. Genetic analysis of 121 patients with MG with no anti-AChR or anti-MuSK antibodies revealed 9 patients with *CHNRA1-*, *CHRNE-*, and *RAPSN*-CMS [30,31,32]. Muscle hypoplasia should be differentiated from congenital myopathies and limb-girdle muscular dystrophies. As stated above, some CMS patients have elevated serum CK levels. CMS should be considered in patients not only with diurnal fluctuation of muscle weakness, but also with day-to-day fluctuation of muscle weakness as well as continuous muscle weakness. Interestingly, 10 patients with *PREPL*-CMS were first considered to be Prader-Willi syndrome [33]. Additionally, two pedigrees with *SYT2*-CMS were initially diagnosed as Charcot-Marie-Tooth disease and distal hereditary motor neuropathy, respectively [9].

In addition, natural and artificial toxins and drugs affect the NMJ signal transmission. For example, (i) a plant toxin, curare, and a snake toxin, α-bungarotoxin, block muscle nicotinic AChR (*CRHNA1, CHRNB1, CHRND, CHRNE*), (ii) a shell toxin, ⍵-conotoxin, blocks N-type calcium channel (*CACNA1B*) at the nerve terminal, (iii) a shell toxin, µ-conotoxin, blocks skeletal muscle sodium channel (Na_V_1.4, *SCN4A*), (iv) a spider toxin, α-latrotoxin, makes a cation-nonselective ion channel pore at the nerve terminal, which allows excessive influx of calcium ions, (v) a bacterial toxin, botulinum, degrades the SNARE complex at the nerve terminal, (vi) chemical weapons, sarin, soman, tabun, and VX, block acetylcholinesterase (AChE), (vii) a pesticide, organophosphate, also blocks AChE, (viii) an antibiotic, aminoglycoside, inhibits calcium uptake at the nerve terminal, (ix) excessive administration of cholinesterase inhibitor (ChEI) blocks AChE, and (x) spores of *Clostridium botulinum* in honey cause infantile botulism, which resembles CMS. The diagnosis of infantile botulism is supported by a self-limited course even when there is no apparent history of honey intake.

Arthrogryposis multiplex congenita (AMC) is caused by defects in more than 320 genes [34]. Pathogenic variants in *CHRNG* show AMC in the lack of myasthenia [35,36,37], and are observed in the largest number of AMC patients [38]. Pathogenic variants in *CHRNA1* [39], *CHRNB1* [39], *CHRND* [39], *RAPSN* [39,40], *SLC18A3* [41], *SNAP25* [15], and *MYO9A* [38] also cause AMC in some patients.

### 2.4. Epidemiology

Analysis of 123 CMS patients in UK showed that the prevalence of CMS under age 18 years largely differ in regions in UK ranging from 2.8 to 14.8 per million with an average of 9.2 per million [42]. This prevalence was about 6 times higher than the prevalence of 1.5 per million of juvenile MG in UK [42]. Similarly, analysis of 22 CMS patients in Brazil [43], 8 CMS patients in Slovenia [44], and 64 CMS patients in Spain [45] showed that the prevalence of CMS under age 18 years were 1.8, 22.2, and 1.8 per million, respectively. All the reports addressed that they underestimated the prevalence because of the presence of undiagnosed CMS patients.

In the 35 causative genes for CMS, pathogenic variants have been frequently observed in genes for AChR ε subunit (*CHRNE*), collagen Q (*COLQ*), rapsyn (*RAPSN*), Dok-7 (*DOK7*), and glutamine--fructose-6-phosphate transaminase 1 (*GFPT1*). Founder effects have been reported in *RAPSN* p.Asn88Lys [46,47,48,49], *DOK7* c.1124_1127dupTGCC [50], *CHRNE* c.1327delG [51], *GMPPB* c.1000G>A (p.Asp334Asn) [52], and *PLEC* c.1_9del (p.Met1_Gly3del) [53].

### 2.5. Inheritance

Autosomal dominant inheritance or hemiallelic pathogenic variants are observed in SCCMS, *SNAP25*-CMS [15,54], *PURA*-CMS [55], and some [9,56,57] but not the other [58,59,60] patients of *SYT2*-CMS. In contrast, other groups of CMS show autosomal recessive inheritance or require pathogenic loss-of-function variants in two alleles. SCCMS is caused by a gain of function missense variant in a single allele, because prolonged AChR channel openings in half of AChRs at the NMJ are sufficient to cause the SCCMS. Both *SYT2*-CMS and *SNAP25*-CMS exhibit LEMS-like CMS, and are likely to be caused by dominant negative effects. PURA-CMS is likely to be caused by a hemiallelic loss-of-function.

### 2.6. Therapeutic Perspectives

Therapeutic strategies for CMS include ChEIs, ephedrine, salbutamol (albuterol in US), amifampridine (3,4-diaminopyridine), quinidine, fluoxetine, and acetazolamide [61]. ChEIs (e.g., pyridostigmine) are effective in many groups of CMS, but are generally ineffective for SCCMS and *DOK7*-CMS. In addition, ChEIs are contraindicated for *COLQ*-CMS [62,63,64] and *LAMB2*-CMS [65], because of severe adverse effects including respiratory arrest in some patients. Although the underlying mechanisms remain unknown, ChEIs sometimes worsen symptoms in *DOK7*-CMS [64,66,67,68,69], *MUSK*-CMS [70], and *LRP4*-CMS [71].

Ephedrine and salbutamol (albuterol) are effective in many groups of CMS including endplate AChR deficiency caused by pathogenic variants in a large number of genes, as well as in *DOK7*-CMS. Sympathetic nerve innervates the NMJ and facilitates the NMJ signal transmission, which is likely to be a pharmaceutical mechanism of the effects of ephedrine and salbutamol (albuterol) [72]. Ephedrine and salbutamol (albuterol) are also effective in some patients with SCCMS and *COLQ*-CMS, which is likely to compensate for AChR deficiency due to endplate myopathy.

Amifampridine blocks voltage-gated potassium channel at the nerve terminal to potentiate the action potential of the motor nerve, and enhances calcium entry into the nerve terminal, which subsequently facilitates release of ACh into the synaptic space. Amifampridine is effective for LEMS-like CMS, which is characterized by compromised SNARE complex. In addition, amifampridine is effective for many groups of CMS except for SCCMS, *AGRN*-CMS, *SLC5A7*-CMS, *SLC25A1*-CMS. Amifampridine is also effective in some patients with *COLQ*-CMS [62,73], although the pharmacological mechanisms remain unknown.

Quinidine [74,75] and fluoxetine [76] ameliorate SCCMS. In a case of *RAPSN*-CMS, fluoxetine prescribed for depression worsened myasthenia [77]. A marked effect of fluoxetine was reported in a case of *COLQ*-CMS [78].

Acetazolamide was effective in two patients of *SCN4A*-CMS [79,80], but was not in another *SCN4A*-CMS [81].

In 27 pregnancies in 16 CMS patients, all patients continued to take drugs. The symptoms were worsened in 63% of the pregnancies but were subsided after delivery [82].

## 3. Physiological Aspects of Neuromuscular Signal Transmission

It is essential to understand the physiology of signal transduction at the NMJ to recognize the pathomechanisms of CMS (Figure 1). The action potential of the spinal motor neuron is delivered to the nerve terminal, and activates the P/Q-type calcium channel (*CACNA1A*). The calcium ions bind to two C2B domains of synaptotagmin 2 (*SYT2*), and activate the SNARE complex to fuse synaptic vesicles containing ACh to the presynaptic membrane [83]. ACh is then released to the 70-nm synaptic space. ACh released from the nerve terminal is hydrolyzed by AChE in synaptic space, and two molecules of ACh that were not captured by AChE bind to AChR to open a cationic ion channel pore. ACh dissociated from AChR is hydrolyzed to choline by AChE. The generated choline in the synaptic space is up taken by high affinity choline transporter (ChT, *SLC5A7*) expressed in the membrane of the nerve terminal [84]. Choline acetyltransferase (ChAT, *CHAT*) in the nerve terminal generates ACh from up taken choline and acetyl-CoA. Vacuolar proton ATPase embedded in the synaptic vesicle generates a proton gradient, which drives an import of ACh into the synaptic vesicle through vesicular acetylcholine transporter (vAChT, *SLC18A3*) [85].

Adult-type AChR is comprised of the α1 (*CHRNA1*), β1 (*CHRNB1*), δ (*CHRND*), and ε (*CHRNE*) subunits (Figure 2). Two α1 subunits and each of β1, δ, and ε subunits make a pentameric AChR (α1_2_β1δε). Embryonic AChR of a α1_2_β1δγ pentamer will be addressed in a section for endplate AChR deficiency. AChR subunits have four transmembrane domains (M1, M2, M3, and M4), and their N- and C-terminals are on the extracellular side. The second transmembrane domain, M2, makes an ion channel pore. The N-terminal regions of AChR subunits make a large extracellular complex, and ACh binds to the interfaces between α1–ε subunits and α1–δ subunits. AChR is a cation-nonselective ion channel that can pass through all cationic ions of Na^+^, Ca^2+^, and Mg^2+^. As Na^+^ is the major cation in the extracellular space, Na^+^ is the major source to make an endplate potential (EPP). The conductance and the burst duration of fetal AChR are ~70% and ~240% of those of adult-type AChR at the human endplate [86].

Depolarization by EPP elicits the opening of skeletal muscle voltage-gated sodium channel (Na_V_1.4, *SCN4A*) to generate a muscle action potential. Na_V_1.4 is expressed throughout the muscle fiber but is enriched at the motor endplate. Muscle action potential goes into the T tubules, where depolarization is sensed by L-type calcium channel (Ca_V_1.1, *CACNA1S*), which constitutes dihydropyridine receptor (DHPR) with other molecules. DHPR is coupled to ryanodine receptor (RyR, *RYR1*), and RyR releases Ca^2+^ from the sarcoplasmic reticulum (SR). Sarcoplasmic Ca^2+^ binds to troponin and displaces tropomyosin that covers the binding sites of actin for the myosin head to contract the muscle fibers.

Many molecules drive AChR clustering at the motor endplate to enable finely tuned signal transmission at the NMJ. Agrin (*AGRN*) released from the nerve terminal binds to LRP4 (*LRP4*) at the motor endplate [88,89]. Two LRP4 molecules bind to two molecules of MuSK (*MUSK*) to make a hetero tetrameric receptor complex. Agrin binds to LRP4 and induces MuSK phosphorylation. MuSK phosphorylation is enhanced by Dok-7 (*DOK7*) [90]. Phosphorylated MuSK then phosphorylates AChR β1 subunit (*CHRNB1*), which binds to submembranous structural protein, rapsyn (*RAPSN*), with a stoichiometry of 2:1 or 1:1 to make AChR clusters at the motor endplate [91]. Rapsyn makes membraneless condensates by phase separation to anchor AChR [92]. Rapsyn binds to β catenin (*CTNNB1*) and chromodomain helicase DNA binding protein 8 (CHD8, *CHD8*) to reinforce the rapsyn network, which is enhanced by Wnt. Wnt binds to the frizzled-like domain of MuSK and also increases β catenin for the reinforcement of rapsyn. LRP4 is a receptor for agrin on the motor endplate, but also mediates a retrograde signal from the motor endplate to the nerve terminal [93,94]. In addition, we reported that Rspo2 [95,96], Fgf18 [97], and Ctgf/Ccn2 [98] are secreted molecules at the NMJ to enhance the agrin-LRP4-MuSK signaling and the formation of the NMJ [99] (Figure 3).

## 4. Thirty-Five Genes in 14 Groups of CMS

CMS are caused by 35 genes, which can be grouped into 14 groups, based on pathomechanisms. Pathomechanisms, clinical features, and therapies are widely variable from category to category.

### 4.1. Endplate AChR Deficiency (CHRNA1, CHRNB1, CHRND, CHRNE, and RAPSN)

#### 4.1.1. Pathomechanisms

Embryonic AChR is composed of α1_2_β1δγ. The embryonic γ subunit is substituted by the adult-type ε subunit after birth to make α1_2_β1δε. The adult-type α1_2_β1δε has a higher conductance and a shorter opening time compared to the embryonic α1_2_β1δγ [86]. Biallelic lack of the ε subunit or (*CHRNE*) in CMS patients can be compensated for by the γ subunit and is not fatal [16,103,104,105]. The expression of γ-AChR is also observed when the expression of ε-AChR is markedly reduced. In contrast, biallelic lack of the α1, β1, and δ subunits (*CHRNA1*, *CHRNB1*, and *CHRND*, respectively) cannot be compensated for by another subunit, and is fatal. Hemiallelic null variants in *CHRNA1*, *CHRNB1*, and *CHRND* are asymptomatic if the other allele has no pathogenic variant, whereas biallelic null variants are observed only in *CHRNE*. Missense variants in genes encoding the α1, β1, δ, and ε subunits (*CHRNA1*, *CHRNB1*, *CHRND*, and *CHRNE*, respectively) that markedly reduce the cell surface expression of AChR cause endplate AChR deficiency [106,107]. Some missense variants in these genes simultaneously cause endplate AChR deficiency, as well as slow or fast channel myasthenic syndrome (SCCMS or FCCMS). Especially, in FCCMS, the reduction in channel opening events to ~50% alone is not pathogenic, but becomes pathogenic when the expression level is also reduced to ~50% [108].

Aberrant splicing of *CHRNA1* causes an unusual form of endplate AChR deficiency. *CHRNA1* has a 75-nt exon P3A between exons 3 and 4 that is unique to human and anthropoids. An exon P3A-skipped *CHRNA1* transcript makes normal AChR, whereas an exon P3A-included *CHRNA1* transcript cannot form AChR. In human skeletal muscle, P3A(+) and P3A(-) transcripts are generated at a ratio of 1:1, although the physiological significance remains unknown. Pathogenic variants in exon P3A and its preceding intron exclusively include exon P3A in pre-mRNA splicing, and the generated P3A(+) transcript causes endplate AChR deficiency [109,110,111].

Pathogenic variants in *RAPSN* encoding rapsyn also cause endplate AChR deficiency. Some missense variants in *RAPSN* retain self-clustering of rapsyn [112], whereas the others do not [113]. Rapsyn phosphorylated by the agrin-LRP4-MuSK pathway forms submembranous network by self-clustering and also activates the E3 ligase activity, which is compromised by a founder variant, p.N88K in *RAPSN* [114]. Four siblings born from consanguineous parents carried pathogenic homovariants in *RAPSN* (c.491G>A, p.R146H) [115]. However, only two of them were affected by CMS, whereas the other two were not. The two affected CMS siblings additionally had homovariants in *AK9*. AK9 encodes one of nine adenylate kinases, and catalyzes a conversion between nucleotide diphosphate and nucleotide triphosphate. The identified variant in *AK9* was a single nucleotide variation (SNV) at 14 nucleotides upstream to the boundary of intron 5 and exon 6. This variant may make a *de novo* translational start site, but no experimental evidence was provided. In addition, as the same *RAPSN* variant was reported in another CMS patient [116], lack of phenotypes in the two siblings without *AK9* remains unknown.

#### 4.1.2. Clinical Features and Therapies

Endplate AChR deficiency caused by pathogenic variants in *CHRNA1*, *CHRNB1*, *CHRND*, and *CHRNE* have been repeatedly reported since 1996 [117]. Frameshifting and nonsense variants are recognized to be pathogenic even without expression studies, but pathogenic missense variants in *CHRNA1*, *CHRNB1*, *CHRND*, or *CHRNE* may cause (i) reduced AChR expression, (ii) SCCMS, or (iii) FCCMS. Except for a dominantly inherited hemiallelic missense variant in a pedigree, which causes SCCMS, the effects of missense variants in *CHRNE* cannot be differentiated without expression studies. Lack of expression studies in most pathogenic variants prevents us from counting the number of patients or original articles with endplate AChR deficiency. However, endplate AChR deficiency and FCCMS have essentially the same clinical features, which are also similar to myasthenia gravis. In contrast to myasthenia gravis, endplate AChR deficiency is present in embryogenesis in patients with pathogenic variants in *CHRNA1*, *CHRNB1*, and *CHRND*, or is present from birth in patients with pathogenic variants in *CHRNE*. These are likely to account for minor facial anomalies, muscle hypoplasia, and lack of diplopia.

*RAPSN*-CMS has been reported in 38 papers [39,40,46,47,49,50,77,112,113,114,115,116,118,119,120,121,122,123,124,125,126,127,128,129,130,131,132,133,134,135,136,137,138,139,140,141,142,143]. A review of 10 patients with *RAPSN*-CMS showed similar clinical features with a neonatal onset, fluctuations of lid ptosis, bulbar signs, neck muscle weakness, mild limb muscle weakness, as well as with episodic worsening of muscle weakness in adults [134]. These symptoms, however, are commonly observed in any groups of CMS, and none is unique to *RAPSN*-CMS.

ChEIs are generally effective for endplate AChR deficiency irrespective of defective genes. We, however, should be aware that excessive administration of ChEIs causes an iatrogenic pathology similar to endplate AChE deficiency due to pathogenic variants in *COLQ* stated below. Ephedrine and salbutamol (albuterol) are also generally effective for endplate AChR deficiency [144]. The effects of adrenergic agonists are likely due to the innervation of sympathetic nerve to the NMJ and the facilitation of the NMJ signal transmission by the sympathetic nerve [72]. In addition, amifampridine is also effective for endplate AChR deficiency [144,145].

### 4.2. Escobar Variant of Multiple Pterygium Syndrome (EVMPS, Escobar Syndrome) (CHRNG) and Lethal Form of Multiple Pterygium Syndrome (LMPS)/Fetal Akinesia Deformation Sequence (FADS) (CHRNA1, CHRND, MUSK, RAPSN, DOK7, and SLC18A3)

#### 4.2.1. Pathomechanisms

Loss-of-function variants of *CHRNG* cause EVMSP (Escobar syndrome) and LMPS, both of which are characterized by arthrogryposis multiplex congenita (AMC) and pterygium likely due to the embryonic immobility [35,36,37]. Escobar syndrome takes a benign non-progressive course. A case of Escobar syndrome with uniparental disomy, in which a specific region of both alleles arises from a single parent, is reported [146]. FADS and LMPS are spectrum disorders [147]. Pathogenic variants of *CHRNA1* [148], *CHRND* [148], *RAPSN* [39,40,147,148], *DOK7* [147,149], and *SLC18A3* [150] also cause LMPS/FADS. The phenotypes are again likely to be caused by embryonic immobility due to defective NMJ signal transmission.

#### 4.2.2. Clinical Features and Therapies

Escobar syndrome has been reported in 101 patients in 72 pedigrees [35,36,38,45,146,151,152,153,154,155,156,157,158,159,160,161]. As the γ subunit is substituted for the ε subunit after birth, patients show no myasthenia or muscle weakness, but is classified into a form of CMS [36]. Some patients with Escobar syndrome have only distal arthrogryposis but no pterygia [37,38,152]. The presence of an incomplete form of Escobar syndrome suggests that pathogenic variants of *CHRNG* are likely to be undetermined in patients with distal arthrogryposis. More than 220 causative genes have been reported in AMC, and pathogenic variants of *CHRNG* are the most common with 6 out 17 pedigrees with AMC [38]. In a report from Spain, 5 out 64 genetically identified CMS patients were Escobar syndrome [45]. Surgical corrections are applied to arthrogryposis.

### 4.3. Slow-Channel CMS (SCCMS) and Fast-Channel CMS (FCCMS) (CHRNA1, CHRNB1, CHRND, and CHRNE)

#### 4.3.1. Pathomechanisms

SCCMS is caused by abnormal prolongation of the opening time of AChR. In contrast, FCCMS is caused by abnormal shortening of the opening time of AChR. Completely oppositive effects on the channel opening times cause defective NMJ signal transmission.

SCCMS is caused by pathogenic missense variants in one allele of *CHRNA1, CHRNB1, CHRND*, and *CHRNE*, encoding the AChR ɑ1, β1, δ, and ε subunits, respectively, and shows autosomal dominant inheritance. A case of autosomal recessive SCCMS was also reported [162,163]. Pathogenic missense variants of SCCMS can be classified into two categories. The first category includes pathogenic missense variants at the extracellular domain especially at the ACh-binding site and at the first transmembrane domain, M1. These variants delay the dissociation of ACh from AChR. The second category includes pathogenic missense variants at the second transmembrane domain, M2, that forms the ion channel pore [164,165]. Three mechanisms may result in defective NMJ signal transmission in SCCMS. First, prolonged openings of AChR ion channel increase the intracellular Na^+^ concentration and depolarize the resting membrane potential, which reduces the amplitude of an endplate potential (EPP) and makes the muscle sodium channel (Na_V_1.4) difficult to sense an EPP. Second, as AChR is a cation-non-selective ion channel, prolonged openings of AChR allow excessive influx of Ca^2+^ ions, that cause endplate myopathy [166]. Ca^2+^ ions constitute 7% of the endplate current of adult-type εAChR, which is higher than fetal γ-AChR. In two pathogenic variants in SCCMS (*CHRNE* p.T284P [164] and *CHRNE* p.V279F [27]), the permeability of Ca^2+^ ions was increased 1.5- to 2.0-folds, which were likely to accelerate endplate myopathy [167]. Third, prolonged openings of AChR desensitize AChR [168]. AChR is physiologically desensitized by prolonged existence of ACh. Desensitized AChR does not respond to ACh and cannot generate EPP anymore. The structure of desensitized *Torpedo* AChR was recently solved [169]. In the desensitized state, the two agonist-binding sites between the ɑ-δ and ɑ-ε subunits of AChR are rotated counterclockwise perpendicular to the membrane, and the structure of the extracellular end of the M4 helix of the ɑ subunit that interfaces with the δ subunit becomes much different.

FCCMS is kinetically opposite to SCCMS. Pathogenic missense variants can be classified into three categories. The first category includes pathogenic missense variants at the extracellular domain including the ACh-binding site of AChR [170]. Interestingly, detailed kinetic analyses reveal that most of the variants at the ACh-binding site [107,170,171,172] affect the ion channel gating rather than ACh-binding to AChR. However, the other variants affect ACh-binding alone [171] or both ACh-binding and the ion channel gating [173]. The second category is comprised of pathogenic variants in the long cytoplasmic loop between the second and third transmembrane domains (M3 and M4). These variants destabilize the open channel state [86,174,175]. The third category is a pathogenic missense variant at the third transmembrane domain (M3) [176]. The enlarged amino acid in the M3 domain displaces the second transmembrane domain (M2) and narrows the ion channel pore made by the M2 domains of five subunits.

#### 4.3.2. Clinical Features and Therapies

SCCMS has been reported in 34 original articles since 1995 [27,31,162,163,165,168,177,178,179,180,181,182,183,184,185,186,187,188,189,190,191,192,193,194,195,196,197,198,199,200,201,202,203,204]. As observed in other autosomal dominant disorders, the onset of SCCMS can be in adolescence or adulthood. Adult-onset patients tend to have mild phenotypes. Weakness of the extensor muscles of the upper limbs is frequently observed in SCCMS, although the underlying mechanisms remain unknown. Weakness of the extensor muscles of the upper limbs is also reported in 10 out of 15 patients with *DOK7*-CMS [66]. R-CMAP in response to a single nerve stimulus is observed in SCCMS, as in *COLQ*-CMS and *PURA*-CMS. A review of 60 SCCMS patients showed that R-CMAP was observed when the opening burst durations of mutant AChR were increased 8.68-fold on average compared to those of wild-type AChR, whereas R-CMAP was not observed when they were increased 3.84-fold on average [198]. Based on knowledge that sodium channel blockers also block AChR ion channel to some extent, shortening of abnormally prolonged AChR channel openings by an antiarrhythmic, quinidine [74] and an selective serotonin reuptake inhibitor (SSRI), fluoxetine [76] was reported by a single channel recordings of SCCMS-AChRs. Indeed, both quinidine [75] and fluoxetine [76] are effective for SCCMS. Amelioration of endplate myopathies in SCCMS requires several months, and immediate effects of quinidine and fluoxetine are not usually observed. A review of 15 SCCMS patients showed that most patients improved by quinidine or fluoxetine, but the effects were not observed for respiratory insufficiency and palpebral ptosis [193]. In their report, 2 out of 6 SCCMS patients with quinidine developed adverse reactions of hypersensitivity reaction and impaired liver function. In 10 SCCMS patients treated with fluoxetine, 7 patients showed clear response, whereas 3 patients either showed adverse effects of serotonergic crisis, lethargy, and hypotension, or could not tolerate higher dose (40 mg/day) [193]. Similarly, in the initial report of fluoxetine for SCCMS, one of two patients had insomnia, drowsiness, and anorexia [76]. A SCCMS patient suffered suicidal ideation soon after commencing fluoxetine, which is well-recognized concern with fluoxetine [205]. Another review of 60 SCCMS patients showed that patients showing good responses to quinidine or fluoxetine started treatment at 11.6 years after the onset of symptoms on average, whereas patients without good responses started treatment at 30.7 years after the onset on average [198]. Although both quinidine and fluoxetine minimally shorten the openings of wild-type AChR, worsening of symptoms was reported in a patient with *RAPSN*-CMS by fluoxetine that was prescribed for depression [77]. ChEIs and amifampridine are ineffective in most SCCMS patients [31,184,188,193], but ChEIs improved the symptoms of a patient with SCCMS [185]. ChEIs presumably enhance the desensitization of AChRs and reduce the number of available AChRs that can respond to ACh. In addition, the effects of ephedrine and salbutamol (albuterol) were reported in SCCMS [193,202,203], as well as in mouse models [206,207].

FCCMS has been reported in 13 original articles since 1996 [64,130,170,170,173,175,208,209,210,211,212,213,214,215]. Biallelic pathogenic missense variants including small indels either cause AChR deficiency or FCCMS. To differentiate the two types of pathologies, microelectrode studies and/or single channel recordings of the biopsied skeletal muscle, and/or single channel recordings of mutant AChR expressed on cultured cells, are required. The unavailability of these techniques is likely to account for the rarity of FCCMS. Indeed, FCCMS has been reported from only three laboratories in the world. Pathogenic variants of FCCMS have been reported in *CHRNA1*, *CHRND*, and *CHRNE*, but not in *CHRNB1*. Although only the β subunit does not contribute to make a ACh-binding site, missense variants in *CHRNB1* can possibly cause FCCMS. Although the differentiation of FCCMS and AChR deficiency is challenging, similar therapies can be applied to both diseases. FCCMS patients respond to ChEIs [2,130,173,200,213], amifampridine [2,213], and salbutamol (albuterol) [216]. Ephedrine is also likely to be effective for FCCMS, but its effect has not been reported. Favorable responses to ChEIs and amifampridine may not necessitate the use of other drugs.

### 4.4. Synaptic CMS (COLQ, LAMB2, and COL13A1)

#### 4.4.1. Pathomechanisms

One, two, and four molecules of AChE enzyme make globular forms of AChE that are named G_1_, G_2_, and G_4_, respectively. In addition, triple helical collagen Q (ColQ) binds to 4, 8, and 12 molecules of AChE and makes asymmetric forms of AChE named A_4_, A_8_, and A_12_, respectively. Asymmetric forms of AChE are enriched at the NMJ. ColQ has three domains. First, the proline-rich attachment domain (PRAD) at the N terminal end is enriched in prolines. The tetrameric forms of AChE bind to PRAD, and three ColQ strands make A_12_ -AChE. Second, the collagen domain in the middle of ColQ has prolines at every three residues like other collagens, and makes a stable triple helical structure. The collagen domain has two regions enriched in positively charged basic amino acids, where heparan sulfate proteoglycans (HSP) including perlecan bind [217]. The two regions are named HSP-binding domain (HSPBD) [218]. Third, the C-terminal domain (CTD) of ColQ is enriched in charged amino acids and cysteines, and makes a globular form. CTD of ColQ binds to MuSK [102,219,220]. Asymmetric forms of AChE are generated in the Golgi apparatus, excreted to the synaptic space, and are anchored to the synaptic basal lamina by binding of ColQ to HSP and MuSK.

Loss-of-function variants of *COLQ* cause endplate AChE deficiency [221,222,223,224,225]. Although the roles of ColQ at the NMJ have been well analyzed, ColQ is also expressed in other tissues including brain, testis, and heart. The roles of ColQ in other tissues, however, remain unknown, and *Colq*-deficient mice show no phenotypes other than endplate AChE deficiency [226,227]. In contrast to *COLQ*, no pathogenic variants have been reported in *ACHE* in any diseases. AChE plays essential roles in the cholinergic synapses in the CNS. Loss-of-function of AChE is thus likely to be fatal in humans. Although there is no relevance to human diseases, p.H322N (rs1799805) in *ACHE* determines the YT blood group [228]. Pathogenic variants of *COLQ* are classified into three categories [224]. First, pathogenic variants in PRAD impair the binding of AChE to PRAD. Second, pathogenic variants in the collagen domain impair the formation of the triple helix. Most of them are nonsense or frameshifting variants. Third, pathogenic variants at CTD impair anchoring of ColQ to the NMJ by inhibiting the binding of ColQ to MuSK [225,229].

Although both endplate AChE deficiency and SCCMS are caused by excessive openings of AChR, the mechanisms of defective NMJ signal transmission are not identical. Two mechanisms are similar between the two diseases. First, depolarization of the resting membrane potential reduces the amplitude of EPP, and small EPP cannot activate the skeletal muscle sodium channel. Second, AChRs are desensitized by the prolonged presence of ACh in endplate AChE deficiency and prolonged openings of AChRs in SCCMS. In contrast to SCCMS, however, endplate myopathy due to excessive influx of Ca^2+^ ions are not observed in endplate AChE deficiency, because the nerve terminal becomes small, and the terminal Schwann cells invaginate into the synaptic space, both of which reduce the number of releasable ACh quanta.

Laminins-221, -421, and -521, all of which include β2-laminin (*LAMB2*), are expressed at the NMJ. Lamins are heterotrimeric extracellular matrix molecules made of ɑ, β, and γ subunits, and are key molecules constituting the synaptic basal lamina [230]. Laminins play a critical role in maintenance of the NMJ, and organization of synaptic vesicle release sites known as active zones. Laminins-221, -421, and -521 are made of ɑ2-, ɑ4-, and ɑ5-laminins, respectively, as well as of β2-, and γ1-laminins. Laminins at the NMJ play essential roles in the juxtaposition of presynaptic and postsynaptic structures and the placement of the terminal Schwann cells at the NMJ. β2-Laminin directly binds to P/Q- and N-type voltage-gated calcium channel (VGCC) [231,232,233], and is essential for the formation and organization of presynaptic active zones [234]. β2-Laminin is also expressed in renal glomeruli and eyes. Pathogenic variants in *LAMB2* cause Pierson syndrome [235] and nephrotic syndrome type 5 [236]. Pierson syndrome is characterized by congenital nephrotic syndrome and a complex maldevelopment of the eye with lens abnormalities, atrophy of the ciliary muscle, corneal changes, and retinal changes. Pathological variants of *LAMB2* were reported in a CMS patient with Pierson syndrome [65]. Ultrastructural analysis of the biopsied muscle showed marked reduction in the size of the nerve terminals, invagination of the synaptic space by the processes of Schwann cells, and moderate simplification of postsynaptic folds. Electrophysiological examinations showed marked reduction in quantal release of ACh from the nerve terminal. *Lamb2*-deficient mice also show similar phenotypes at the NMJ [237].

Collagen 13ɑ1 (*COL13A1*) enriched at the NMJ has a single transmembrane domain and plays an essential role in the maturation and maintenance of AChR at the NMJ [238]. A frameshifting variant of *COL13A1* causes CMS [239]. Introduction of the pathogenic variants into C2C12 myotubes reduced AChR clustering [239]. *Col13a1*-deficient mice show abnormal formation of the NMJ [238,240], as well as craniofacial malformations and a reduction in cortical bone mass in aged mice [241].

#### 4.4.2. Clinical Features and Therapies

*COLQ*-CMS has been reported in 30 original articles since 1998 [62,73,78,140,141,221,222,225,242,243,244,245,246,247,248,249,250,251,252,253,254,255,256,257,258,259,260,261,262,263]. Interestingly, a grandmother and a father of two siblings with *COLQ*-CMS carried a heterozygous truncation variant of *COLQ*, and showed congenital ptosis [246]. Although the presence of a pathogenic variant on another allele could not be excluded, a heterozygous variant of *COLQ* might have caused ptosis. Initial symptoms of *COLQ*-CMS are mostly ophthalmoplegia, respiratory insufficiency, and weak crying at birth. Follow-up of 15 patients with *COLQ*-CMS aged 3 to 48 years for up to 10 years showed that 80% of patients were still ambulant and 87% had no respiratory difficulties [253]. A report of 22 patients with *COLQ*-CMS indicated proximal dominant muscle weakness that was characteristic of limb-girdle-type myasthenia as in *DOK7-*CMS [247]. Fluctuating scoliosis due to truncal muscle weakness is later changed to severe scoliosis, which is uniquely observed in *COLQ*-CMS and *DOK7-*CMS [249]. Palpebral ptosis and external ophthalmoplegia are observed in about half of *COLQ*-CMS patients [247,253]. Diurnal fluctuation and progression of muscle weakness are observed in about half of the patients [253]. Delayed pupillary response is characteristic of *COLQ*-CMS but is observed in only 25% of the patients [247]. Repetitive CMAP, which is also observed in SCCMS and *PURA*-CMS, is observed in about half of the patients [253]. In *COLQ*-CMS, globular forms of AChE and butyrylcholinesterase at the motor endplate catalyze ACh, and blocking of these enzymes by ChEIs can sometimes cause respiratory arrest [62,63,64]. ChEIs had no long-term effects in 22 patients with *COLQ*-CMS but showed short-time effects in 4 patients [247]. Ephedrine and salbutamol (albuterol) are effective for *COLQ*-CMS [264,265,266]. Especially, in two patients, ephedrine showed marked effects [264], although the underlying mechanisms remain unknown. The effects of amifampridine are also reported [62,73], but the mechanisms again remain elusive. The effect of fluoxetine that is used for SCCMS was also reported in a patient with *COLQ*-CMS [78]. Although fluoxetine minimally shortens the channel opening time of wild-type AChR [74], a slight reduction in AChR openings was likely to be sufficient for the patient.

*LAMB2*-CMS was reported in a 20-year-old female with Pierson syndrome in 2009 [65]. The patient had repeated respiratory distress since birth, miosis, and severe proteinuria. Development of motor functions were delayed, but proteinuria was improved by rental transplantation at age 7 years. Palpebral ptosis, external ophthalmoplegia, and proximal muscle weakness were noted. RNS reduced CMAP by 24%. ChEI worsened her muscle weakness, and respiratory support was required. Ephedrine was effective.

*COL13A1*-CMS has been reported in 41 patients in 19 pedigrees since 2015 [140,141,239,267,268,269]. All patients developed respiratory distress and weak sucking at birth. In addition to severe palpebral ptosis and mild external ophthalmoplegia, the patients showed facial, bulbar, respiratory, and truncal muscle weakness. Compared to the trunk muscles, limb muscles are spared. ChEIs are ineffective [239,267,268], but salbutamol (albuterol) [239,267,268] and amifampridine [268] are effective.

### 4.5. Sodium Channel CMS (SCN4A)

#### 4.5.1. Pathomechanisms

Loss-of-function variants of *SCN4A* cause CMS [79,81,270], whereas gain-of-function variants of *SCN4A* cause hyperkalemic periodic paralysis [271], hypokalemic periodic paralysis [271], potassium-aggravated myotonia congenita [272], and paramyotonia congenita [273]. Loss-of-function variants of *SCN4A* in CMS shift the fast inactive curve toward hyperpolarized states and make Na_V_1.4 inactive even at the resting membrane potential. Na_V_1.4 ion channel opens in response to the first depolarization stimulus, but not to the second or later depolarization stimuli because of accelerated transition to a fast inactive state. This also causes decremental CMAP response to RNS. In contrast to *SCN4A*-CMS, gain-of-function variants in hyperkalemic periodic paralysis, hypokalemic periodic paralysis, potassium-aggravated myotonia congenita, and paramyotonia congenita shift the fast inactivation curve toward depolarized states. This allows repeated openings of Na_V_1.4 or allows leakage of Na^+^ even in a closed state. In *SCN4A*-CMS and in some CMS patients with defective recycling of ACh (*CHAT*-CMS, *SLC18A3*-CMS, *SLC5A7*-CMS, and *PREPL*-CMS), a high-frequency nerve stimulation is required to elicit a decremental CMAP response, and episodic muscle weakness is observed.

#### 4.5.2. Clinical Features and Therapies

*SCN4A*-CMS has been reported in 6 patients since 2003 [79,80,81,140,270,274]. *SCN4A*-CMS shows frequent episodes of respiratory arrest, bulbar paralysis, and muscle weakness that persist 30 to 60 min. In the intermittent phase, mild facial, truncal, and limb muscle weakness, as well as external ophthalmoplegia, are observed. Analysis of 278 patients with sudden infantile death syndrome (SIDS) revealed 4 patients with *SCN4A*-CMS [5]. ChEIs are either effective [79,142,274] or ineffective [270]. In addition, a SCN4A-CMS patient showed marked cholinergic adverse effects with a small amount of ChEI [80]. Salbutamol (albuterol) was effective in a single patient [142]. Similarly, acetazolamide was either effective [79,80] or ineffective [81] to prevent episodic muscle weakness.

### 4.6. Defective AChR Clustering (AGRN, LRP4, MUSK, and DOK7)

#### 4.6.1. Pathomechanisms

Agrin (*AGRN*) is a large molecule secreted from the nerve terminal with a molecular weight of ~200 kDa, and carries binding domains for laminins, neural cell adhesion molecule (NCAM), α-dystroglycan, and LRP4. Functionally characterized pathogenic variants of *AGRN* invariably impair AChR clustering. However, three pathologies exist depending on the affected domains: (i) impairment of MuSK phosphorylation, (ii) accelerated degradation of agrin, (iii) impaired anchoring of agrin to the NMJ [275].

The third β propeller domain of LRP4 binds to agrin. Pathogenic variants in this domain in CMS impair binding of LRP4 to agrin and MuSK, reduce MuSK phosphorylation, and compromise AChR clustering [71]. Pathogenic variants in this domain are also reported in sclerosteosis type 2 (SOST2), which is characterized by cortical hyperostosis [276]. CMS variants affect agrin-LRP4-MuSK signaling but not Wnt signaling, whereas SOST2 variants affect Wnt signaling but not agrin-LRP4-MuSK signaling. Analysis of additional artificial variants revealed that variants at the periphery of the third β propeller domain exclusively affect agrin-LRP4-MuSK signaling, whereas variants at the center of the domain exclusively affect Wnt signaling [71]. Pathogenic variants of the other domains of *LRP4* are also reported in another bone disorder, Cenani-Lenz syndactyly syndrome [277]. Thus, pathogenic variants of *LRP4* either affect agrin-LRP4-MuSK signaling or Wnt signaling.

Pathogenic variants of *MUSK* either reduces cell membrane expression of MuSK without affecting agrin-mediated phosphorylation of MuSK [278], or markedly reduces MuSK phosphorylation and AChR clustering [279].

More than 70 missense, truncation, and splicing pathogenic variants have been reported in *DOK7* in CMS [280,281,282,283]. Thirteen missense variants have been functionally characterized, and all reduce the phosphorylation of MuSK and AChR β1 subunit [280,282,283,284]. One missense variant in the pleckstrin homology domain markedly reduces DOK7 expression by generating aggresome at the juxtanuclear region [284].

#### 4.6.2. Clinical Features and Therapies

*AGRN*-CMS has been reported in 13 original articles since 2009 [6,141,142,274,275,285,286,287,288,289,290,291,292]. Two patients with *AGRN*-CMS reported in 2009 were 42-year-old female and 36-year-old male in a single pedigree, who had mild limb muscle weakness and unilateral ptosis since childhood [285]. ChEIs and amifampridine were ineffective. Most *AGRN*-CMS patients similarly develop muscle weakness since childhood, and the symptoms range from mild muscle weakness in lower limbs to severe muscle weakness that requires respiratory support. Again, ChEIs and amifampridine are ineffective or mildly effective. On the other hand, salbutamol (albuterol) was effective in 10 out of 12 *AGRN*-CMS patients [287]. Similarly, ephedrine was effective in a single patient [288]. Biallelic null variants of *AGRN* caused FADS and gave rise to stillbirth at 30 weeks of gestation [293]. In addition, analysis of 262 patients with autism spectrum disorder (ASD) revealed hemiallelic null variants of *AGRN* [294]. However, as null variants of *AGRN* are observed in asymptomatic parents of *AGRN*-CMS patients, other genetic or environmental factors are likely to be required to be associated with ASD. In addition, pathogenic *AGRN* variants were identified in hereditary motor neuropathy, in whom jitters by SFEMG were increased [295].

*MUSK*-CMS has been reported in 15 original articles since 2004 [44,70,141,278,279,296,297,298,299,300,301,302,303,304,305]. A review of 15 *MUSK*-CMS patients showed that the ages of onset were from birth to 8 years, and most patients had proximal muscle weakness, palpebral ptosis, external ophthalmoplegia, facial weakness, bulbar palsy, and truncal muscle weakness [70]. About half of the patients required respiratory support for respiratory insufficiency. ChEIs were effective or worsened muscle weakness, and amifampridine and salbutamol (albuterol) were mildly or markedly effective [70]. In addition, 19 patients with FADS due to pathogenic variants of *MUSK* were reported [306,307].

*LPR4*-CMS with compound heterozygous variants was reported in a single patient in 2014 [71]. The patient had respiratory distress at birth, and was dependent on a respirator up to age 6 years. Evaluations at ages 9 and 14 years showed mild external ophthalmoplegia and severe muscle weakness. ChEIs worsened muscle weakness.

*DOK7-*CMS has been reported in 34 original articles since 2006 [43,45,50,64,66,67,68,69,139,140,141,142,147,280,281,282,283,308,309,310,311,312,313,314,315,316,317,318,319,320,321,322,323,324]. A review of 15 patients with *DOK7-*CMS showed that the ages of onset were mostly from birth to infancy, and the oldest age of onset was 13 years [66]. All patients showed proximal and truncal muscle weakness, and scoliosis was frequently observed. In addition, distal muscle weakness, especially finger extensors, was observed in 12 patients. Similar, weakness of finger extensors is also observed in SCCMS. Muscle hypoplasia was present in about half of the patients; palpebral ptosis and external ophthalmoplegia in 11 patients; and facial and bulbar muscle weakness in 8 to 9 patients. *DOK7-*CMS is recognized as limb-girdle CMS, but ocular, facial, and bulbar weakness is frequently observed. The diagnosis of myasthenia gravis was erroneously given to 4 out of 15 patients, and others were diagnosed as congenital myopathy, metabolic myopathy, or mitochondrial myopathy. Although the causal relation remains unknown, siblings of *DOK7-*CMS had mitral valve insufficiency [322]. A total of six CMS patients were heterozygous for a truncation variant of *DOK7* without any pathogenic variants on another allele [281,325]. Although the presence of a pathogenic variant on another allele could not be excluded, a heterozygous variant of *DOK7* might cause CMS when unidentified genetic and/or environmental factor(s) coexisted. More such heterozygous patients may exist, but may be underestimated due to possible publication bias. The effects of ephedrine and salbutamol (albuterol) for *DOK7-*CMS have been repeatedly reported [64,66,67,68,69,308,310,315,316,317,321,324]. In addition the effect of a patch of tulobuterol, a β2 agonist, was reported in a case with *DOK7-*CMS [326]. On the other hand, ChEIs are ineffective or worsen muscle weakness [64,66,67,68,69]. The effect of amifampridine was also reported [319]. Fluoxetine was effective in a patient with *DOK7*-CMS, who was misdiagnosed as SCCMS [327]. Administration of anti-DOK7 antibody that stimulated DOK7 was effective for a mouse model of *DOK7*-CMS carrying a pathogenic variant of the patient [328]. Although the mechanisms are unknown, a *DOK7*-CMS patient was responsive to steroid for 40 to 50 years [329]. In addition, 4 patients with FADS due to pathogenic variants of *DOK7* were reported [147,149].

### 4.7. CMS Caused by Defective Structural Molecule at the NMJ (PLEC)

#### 4.7.1. Pathomechanisms

Plectin is a 500-kD intermediate filament-binding protein that provides mechanical strength by acting as a crosslinking element of the cytoskeleton. In the skeletal muscle, plectin is expressed in sarcolemma and Z band. In the skin, plectin makes hemidesmosome. Pathogenic variants of *PLEC* cause epidermolysis bullosa simplex (EBS) [330] and autosomal recessive limb-girdle muscular dystrophy 17 (LGMD17) [331]. In patients with EBS and LGMD17, endplate AChR deficiency was reported [332,333,334]. A homozygous 9-bp deletion of the translational start site of *PLEC* caused LGMD17 in 3 patients in 3 pedigrees in Turkey without any evaluation of the NMJ [331]. The same variant, however, caused LGMD17 and endplate AChR deficiency in 4 patients in 4 pedigrees in Turkey [53], indicating that myasthenic symptoms might be masked by muscular dystrophy. Plectin is highly expressed at the NMJ, connects desmin and dystrophin-dystroglycan complex, binds to rapsyn-AChR complex, and stabilizes the NMJ structure [335]. Indeed, ultrastructural analyses show destruction and remodeling of the endplate [332].

#### 4.7.2. Clinical Features and Therapies

*PLEC*-CMS has been reported in 22 patients since 1999 [53,274,332,334,336,337,338,339]. Although LGMD17 and CMS are always present, EBS may [332,334,336,338] or may not [53,337] be present. Some patients also show mild EBS [338]. A review of 117 *PLEC*-EBS patients carrying pathogenic variants in *PELC* showed that 14 patients also had CMS [339]. However, the authors observed the presence of CMS in 7 out of 15 patients in their own cohort of *PLEC*-EBS [339], indicating that CMS was likely to be underdiagnosed and that the prevalence of CMS was higher than reported. The prevalence of muscular dystrophy in *PLEC*-EBS was also high. The onsets of *PLEC*-CMS range from early childhood to age 26 years, and patients show limb muscle weakness, swallowing difficulty, respiratory insufficiency, palpebral ptosis, external ophthalmoplegia [53,332,334,338]. As in most other patients with CMS, low-frequency RNS elicit decremental CMAP responses. ChEIs were ineffective in 3 patients [334], and effective in 3 other patients [338]. A combination of ChEIs and salbutamol (albuterol) was effective in 4 patients [53]. Amifampridine was effective in a case [338], and was ineffective in 2 other patients [334,338]. In addition, a case of *PLEC*-*CHRNE*-CMS who had both biallelic insertion of 36 bp in *PLEC* and biallelic frameshift in *CHRNE* showed EBS and CMS [333]. ChEIs and ephedrine were mildly effective for this case.

### 4.8. CMS Caused by Defective Recycling of ACh (CHAT, SLC18A3, SLC5A7, and PREPL)

#### 4.8.1. Pathomechanisms

Choline acetyltransferase (ChAT, *CHAT*) synthesizes ACh from choline and acetyl-CoA at the nerve terminal. Vesicular acetylcholine transporter (vAChT, *SLC18A3*) transports synthesized ACh to the synaptic vesicle. *SLC18A3* is encoded within the first intron of *CHAT*. This nested gene structure is conserved from *C. elegans*. Loss-of-function variants of *CHAT* cause CMS with episodic apnea [340,341]. ChAT is also expressed at the cholinergic synapses in the CNS, and developmental delay that is observed in about half of *CHAT*-CMS patients can be accounted for either by defects in the cholinergic synapse in the CNS or by hypoxia due to episodic apnea. Parents of *CHAT*-CMS who carry a null variant in a single allele are asymptomatic, whereas no *CHAT*-CMS patients carry biallelic null variants. Thus, the reduction in the enzymatic activity of ChAT to 30-50% is predicted to cause CMS, whereas ChAT activities lower than 30% are lethal and more than 50% are asymptomatic [340,341].

Loss-of-function variants of *SLC18A3* also cause CMS. Although the disease mechanisms have not been dissected in detail, failure to pack resynthesized ACh into synaptic vesicles is likely to be the cause of CMS.

A hemiallelic large scale DNA rearrangement at 10q11.2 including *CHAT* and *SLC18A3* was observed in 41 patients with autism, developmental delay and/or intellectual disability, and multiple congenital malformations [342]. Muscle hypotonus, palpebral ptosis, and sleep apnea in these patients may be caused by haploinsufficiency of *CHAT* and *SLC18A3.* However, as stated above, as hemiallelic null variants are symptomatic in parents of *CHAT*-CMS, either complete lack of vAChT (*SLC18A3*) that is intact in *CHAT*-CMS or an unidentified pathogenic variant on the other allele may cause the disease. Indeed, in two patients with CMS, a hemiallelic large scale deletion at 10q11.2 region was unmasked by a pathogenic splicing variant in *CHAT*, or a pathogenic missense variant in *SLC18A* [343].

High affinity choline transporter (ChT, *SLC5A7*) expressed at the nerve terminal membrane uptakes choline to the nerve terminal. ChT is a homo-oligomeric membrane transporter. A hemiallelic frameshifting variant of *SLC5A7* cause autosomal dominant distal hereditary motor neuropathy type VIIA (DHMN7A) [344,345]. DHMN7A is characterized by teen-age onset progressive distal limb muscle weakness and amyotrophy with vocal paralysis. Later, recessive pathogenic variants of *SLC5A7* were reported to cause CMS [41]. Expression studies of cultured cells showed that dominantly inherited variants of *SLC5A7* have dominant-negative effects [344], whereas recessively inherited variants have loss-of-function effects [41,346], on choline uptake by ChT. Dominantly inherited variants of *SLC5A7* are likely to inhibit the formation of homo-oligomers of ChT, whereas recessively inherited variants do not. However, it remains unknown why similar reductions of the ChT activity give rise to two different phenotypes of DHMN7A and CMS. Mice deficient for *Slc5a7* dies in a few minutes after birth probably due to respiratory failure [347], and spinal motor neuron-specific rescue of *Slc5a7* prolonged the knockout move and breath for ~24 h after birth [348]. Hemizygous knockout of *Slc5a7* in mice decreased cardiac ACh and showed diminished parasympathetic heart effects with basal resting tachycardia [349], which, however, has not been documented in patients with *SLC5A7*-CMS.

Prolyl endopeptidase-like (*PREPL*) is one of serine peptidases, and its physiological substrate is unknown. The *SLC3A1* gene encoding the cystine, dibasic, and neutral amino acid transporter and the *PREPL* gene are encoded on the opposite strands each other and have an overlap at their 3′ ends. Deletion of both genes cause hypotonia-cystinuria syndrome (HCS) [350]. Deletion of *PREPL* causes muscle hypotonia [351], whereas deletion of *SLC3A1* causes cystinuria [352]. Loss-of-function variants of *PREPL* do not cause endplate AChR deficiency, but inhibit refilling of ACh to synaptic vesicles, reduce the number of releasable ACh quanta, and decrease the probability of vesicular release [351].

#### 4.8.2. Clinical Features and Therapies

*CHAT*-CMS has been reported in 19 original articles since 2001 [45,140,141,142,340,343,353,354,355,356,357,358,359,360,361,362,363,364]. A follow-up study of 11 patients with *CHAT*-CMS for maximum of 12 years showed two forms of disease [358]. It can present in neonates with episodic apnea, respiratory distress, swallowing difficulty, and limb muscle weakness. It can also start in infancy with episodic apnea, and mild limb muscle weakness. The milder infantile form may show progressive muscle weakness with wheelchair dependency. Episodic apnea is frequently misdiagnosed as epilepsy [358]. Episodic apnea, however, is observed in other groups of CMS, and is not unique to *CHAT*-CMS. Similar to *SCN4A*-CMS, about half of *CHAT*-CMS patients show no decremental response to low-frequency RNS, and require high-frequency RNS at 10 Hz or more. A respiratory monitor is required for neonatal and infantile episodic apnea. ChEIs and amifampridine are effective in most patients with *CHAT*-CMS [340,341].

*SLC18A3*-CMS has been reported in seven patients in six pedigrees since 2016 [343,365,366,367]. Patients show severe muscle weakness at birth, muscle hypotonia, arthrogryposis, and respiratory distress. Although not observed in all the patients, palpebral ptosis, external ophthalmoplegia, and episodic apnea are also observed [343,365,367]. In addition, FADS was reported in two patients with biallelic nonsense variants of *SLC18A3* [150]. RNS was documented in three patients: two showed decremental CMAP in response to low-frequency RNS [365,367], whereas one showed decremental CMAP only after isometric muscle contractions as observed in *SCNA4*-CMS and *CHAT*-CMS [365]. ChEIs [365,366,367], ephedrine [365,367], and amifampridine [365,367] are effective for *SLC18A3*-CMS.

*SLC5A7-*CMS has been reported in 12 patients in 10 pedigrees since 2016 [41,346,368,369]. Typical clinical features include neonatal-onset episodic apnea, muscle hypotonia, muscle weakness, and weak crying. Some patients have arthrogryposis and congenital malformations and die in infancy, and some patients show developmental delay. Progressive brain atrophy was reported in a case with *SLC5A7-*CMS, which was likely due to repeated apneustic attacks [346]. In addition, repeated intestinal perforations were reported in two patients of *SLC5A7-*CMS in a single pedigree [346]. Among 6 patients with *SLC5A7*-CMS, in whom RNS results were documented, 5 patients showed decremental CMAP to low-frequency RNS [41,346,368], and a single patient showed decremental CMAP only after RNS at 20 Hz for 10 sec [41,365]. ChEIs are effective [41,346,368], and ephedrine has an additional effect [346]. Amifampridine is ineffective [346].

*PREPL-*CMS without pathogenic variants *SLC3A1* has been reported in 11 patients since 2014 [33,351,370,371,372,373,374,375,376]. Similarly, *PREPL-*CMS with *SLC3A1* deletion, the diagnosis of which is HCS, has been reported in 7 patients since 2014 [33,351]. *PREPL*-CMS is characterized by fluctuating muscle weakness and feeding difficulty since birth, and sometimes requires respiratory support. Patients show palpebral ptosis, nasal voice, swallowing difficulty and facial muscle weakness, and sometimes proximal limb muscle weakness. Intelligence is normal or slightly affected. CMAP decreases with low-frequency RNS [371,376]. In a patient with *PREPL*-CMS, however, low-frequency RNS elicited decremental CMAP only after RNS at 20 Hz for 2 min [376]. Ten patients with *PREPL*-CMS including HCS were initially diagnosed as Prader-Willi syndrome [33]. ChEIs are effective [35,351,372,374]. The first *PREPL*-CMS patient could discontinue ChEI at age 12 months, although muscle weakness was still present [351].

### 4.9. Lambert-Eaton Myasthenic Syndrome (LEMS)-Like CMS (SYT2, SNAP25, VAMP1, UNC13A, RPH3A, and LAMA5)

#### 4.9.1. Pathomechanisms

Synaptotagmin 2 (*SYT2*) senses Ca^2+^ ions entering into the nerve terminal through P/Q-type calcium channel and triggers the formation of the SNARE complex that releases ACh in synaptic vesicles to the synaptic space. Hemiallelic pathogenic variants were identified in *SYT2* in patients with CMS resembling LEMS [56,57]. Functional analysis with *Drosophila* showed that the variants indeed affected the release of synaptic vesicles [56,57].

The SNARE complex is made of SNAP25, syntaxin, and synaptobrevin (vesicle-associated membrane protein 1, VAMP1). Hemiallelic *de novo* loss-of-function missense variants of *SNAP25* cause CMS with developmental delay and ataxia [15]. *SNAP25* has two splicing isoforms: *SNAP25A* transcript includes 118-bp exon 5A, whereas *SNAP25B* transcript includes 118-bp exon 5B. Embryonic *SNAP25A* is switched to adult-type *SNAP25B* after birth. Pathogenic variants in exon 5B encoding *SNAP25B* cause CMS [15]. t-SNARE liposome containing a variant SNAP25B failed to properly fuse to v-SNARE liposome induced by calcium ions. In addition, bovine chromaffin cells expressing a variant SNAP5B showed compromised exocytosis in response to depolarization.

Another component of the SNARE complex, syntaxin 1, is bent in the middle and is in a closed conformation at rest. Munc18-1 stabilizes syntaxin 1 in a closed state. In response to the entry of Ca^2+^ ions to the nerve terminal, Munc13-1 (*UNC13A*) binds to syntaxin 1 and displaces Munc18-1, which stabilizes the open conformation of syntaxin 1 [377]. Biallelic truncation variants of *UNC13A* caused severe muscle weakness at birth, microcephaly, hypoplastic corpus callosum, enhanced excitation of cerebral cortex [11]. A microelectrode study of biopsied skeletal muscle showed that *UNC13A*-CMS decreased the quantal contents of synaptic vesicles but spared the release probability of synaptic vesicles. In contrast to *UNC13A*-CMS caused by biallelic truncation variants, hemiallelic pathogenic missense variants of *UNC13A* do not cause CMS but cause dyskinesia, developmental delay, and autism [378].

Biallelic loss-of-function variant of *VAMP1* encoding another component of the SNARE complex, synaptobrevin1, cause a neonatal onset CMS [379,380,381]. *Vamp1*-deficient mice showed marked shrinkage of motor endplate and reduction in endplate potentials, which is electrophysiologically similar to LEMS [379].

Rabphilin 3a (*RPH3A*) is an effector of a Ras superfamily molecule, Rab3A, and binds to Rab3A at the nerve terminal. In addition, Rabphilin 3a binds to SNAP25 and 14-3-3 proteins. In *Drosophila*, 14-3-3ζ binds to and regulates potassium channel at the nerve terminal of the NMJ [382]. Although *Rph3a* knockout shows no phenotypes in mice [383] or *Drosophila* [384], microinjection of rabphilin 3a into the squid giant axon suppresses release of synaptic vesicles [385]. Two pathogenic missense variants identified in a patient with *RPH3A*-CMS reduced the binding of rabphilin 3a to 14-3-3, but not to Rab3A or SNAP-25 [12].

Laminin α5 (*LAMA5*) is highly expressed at the NMJ. Knockout of *Lama5* results in embryonic lethality in mice [386], whereas muscle-specific knockout of both *Lama4* and *Lama5* markedly affect the postsynaptic structure [387]. Muscle-specific knockout of *Lama5* alone does not show any motor deficit, but differentiation of the nerve terminal is compromised, and the nerve terminal is juxtaposed to only a part of motor endplate [387]. In *LAMA5*-CMS, quantal release of ACh is markedly reduced. In addition, although the junctional folds of the motor endplate are spared, the motor endplate is not covered by the nerve terminal or covered by a small nerve terminal [13]. Synaptic vesicle glycoprotein 2A (SV2A) binds to synaptotagmin, and pathogenic variants of *LAMA5* impairs binding to SV2A [13].

Some patients with *SYT2*-CMS [9,56,57,60] and all the two patients with *SNAP25*-CMS [15,54] are caused by hemiallelic pathogenic variants, whereas the other LEMS-like CMS requires biallelic pathogenic variants.

#### 4.9.2. Clinical Features and Therapies

*SYT2*-CMS has been reported in 6 original articles since 2014 [9,56,57,58,59,60]. Hemiallelic pathogenic variants caused *SYT2*-CMS in 15 patients in 4 pedigrees [9,56,57,60], whereas biallelic premature termination codons caused *SYT2*-CMS in 9 patients in 7 pedigrees [58,59,60]. Patients with hemiallelic variants show childhood onset, whereas those with biallelic variants show severe phenotypes mostly with neonatal or infantile onset. Ten patients in 2 pedigrees in the first report had myasthenia but had no ptosis or external ophthalmoplegia [9,56]. Low-frequency RNS elicited decremental CMAP responses, and short maximal voluntary contractions enhanced CMAP amplitudes as observed in LEMS. They were initially diagnosed as Charcot-Marie-Tooth disease or hereditary distal motor neuropathy [9]. ChEIs [58,59] and amifampridine [9,59] were effective for *SYT2*-CMS. Amifampridine was more effective than ChEIs [9]. Salbutamol (albuterol) had no effect [59].

*SNAP25*-CMS was reported in a single patient in 2014 [15] and another in 2022 [54]. Both were caused by hemiallelic pathogenic variants. *SNAP25*-CMS shows severe muscle hypotonia and muscle weakness at birth, and arthrogryposis multiplex. A case in 2014 could walk with a walker at age 7 years, and sometimes had palpebral ptosis [15]. In a case in 2014, low-frequency RNS caused a decremental CMAP, but high-frequency RNS was not performed [15]. In a case in 2022, no RNS studies were performed [54]. ChEI was ineffective, but amifampridine was effective [15]. A case in 2022 died at age 6 days due to respiratory distress [54].

*UNC13A*-CMS was reported in a single patient in 2016 [11]. *UNC13A*-CMS showed severe muscle hypotonia and muscle weakness at birth, and microcephaly and hypoplastic corpus callosum. Low- and high-frequency RNS showed decremental and incremental CMAP responses, respectively. EEG showed sharp waves, but no epileptic attacks were noted. ChEI and amifampridine improved decremental CMAP in response to RNS, but minimally improved clinical symptoms. The patient died at age 50 months due to respiratory failure.

*VAMP1*-CMS has been reported in 9 patients in 7 pedigrees since 2017 [10,379,380,381]. Patients had muscle hypotonia, muscle weakness, and myasthenia since birth [10,379,381] or age 6 months [380]. CMAPs decremented and incremented in response to low- and high-frequency RNS, respectively [10,379]. External ophthalmoplegia and bulbar palsy were also noted [10,379,381]. ChEIs were effective for *VAMP1*-CMS [10,379,380,381]. Amifampridine has not been administered to *VAMP1*-CMS, and the effects remain unknown.

*RPH3A*-CMS was reported in a 11-year-old female [12]. She had limb muscle weakness, nasal voice, and intolerance to exercises since age 3 years. She had learning disabilities. No palpebral ptosis or external ophthalmoplegia was noted. She had mild proximal limb muscle weakness and cervical muscle weakness. Although the association to CMS is unknown, repeated abdominal pain and hyperglycemia are also noted. CMAP amplitudes were not decreased at 2 Hz RNS, but were increased at 30 Hz RNS, as observed in LEMS. Salbutamol (albuterol) was effective, and other drugs were not used.

*LAMA5*-CMS was reported in a single patient in 2017 [13]. The patient was noted with weak cry and was dependent on a respirator. A brother died of muscle weakness, but details were unknown. Minor facial anomalies were also noted. Low-frequency RNS decreased CMAP by maximum 55%. Low-frequency RNS after maximum muscle contraction for 30 sec increased CMAP to 250%. Coadministration of ChEI and amifampridine was effective.

### 4.10. Glycosylation-Deficient CMS (GFPT1, DPAGT1, ALG2, ALG14, GMPPB)

#### 4.10.1. Pathomechanisms

Glutamine-fructose-6-phosphate transaminase 1 (GFPT1) is a rate-limiting enzyme to produce uridine diphosphate N-acetylglucosamine (UDP-GlcNAc) that is an essential source for N- and O-glycosylations (Figure 4). On the other hand, dolichyl-phosphate N-acetylglucosamine phosphotransferase 1 (DPAGT1) and UDP-N-acetylglucosaminyltransferase subunit (asparagine-linked glycosylation 14 homolog, ALG14) work on the first two steps of adding GlcNAc to dolichyl phosphate in N-glycosylation. Pathogenic variants of *DPAGT1* [388] and *ALG2* [389] were reported in congenital disorder of glycosylation Ij (CDG Ij) with infantile spasms, developmental delay, microcephaly, and finger malformations. Muscle hypotonia and muscle weakness are documented in CDGs, and some CDGs may have CMS. Alpha-1,3/1,6-mannosyltransferase (asparagine-linked glycosylation 2 homolog, ALG2) add mannose in N-glycosylation pathway. GDP-mannose pyrophosphorylase B (GMPPB) makes GDP-mannose from mannose-1-phosphate and GTP. GMPPB is essential for N- and O-mannosylations. Expression of pathogenic variants of GFPT1 causes abnormal structures of muscle fibers and NMJ in zebra fish [390]. In C2C12 myotubes, knockdown of *Gfpt1* [391] and *Alg14* [24] markedly reduced cell surface expression of AChR. Although pathogenic variants of these genes cause AChR deficiency in cultured cells, the detailed mechanisms remain to be elucidated.

#### 4.10.2. Clinical Features and Therapies

*GFPT1*-CMS has been reported in 17 original articles since 2011 [18,19,20,45,135,140,274,390,392,393,394]. *DPAGT1*-CMS [21,395,396,397,398] and *GMPPB*-CMS [25,29,140,141,274,399,400] have been reported in five and nine original articles, respectively. *ALG2*-CMS has been reported in 9 patients in 4 pedigrees since 2013 [24,401]. *ALG14*-CMS has been reported in 12 patients in 7 pedigrees since 2013 [24,402,403,404,405]. In muscle biopsy of a patient with *GFPT1*-CMS, glycogen storage was observed, and glycogen storage disease was considered [406]. In 12 patients with *ALG14*-CMS, 10 patients had epilepsy [402,403,404,405], and 2 had severe intellectual disability [403].

Pathogenic variants of *GFPT1* [390], *DPAGT1* [21], *ALG2* [24], *ALG14* [24] cause limb-girdle CMS with tubular aggregates. Three patients with *GFPT1*-CMS had rimmed vacuoles in skeletal muscle [19], and 2 patients with *GFPT1*-CMS had myofibrillar myopathy with deposition of desmin [394]. Pathogenic variants of *GMPPB* also cause limb-girdle CMS, but no tubular aggregates [25]. Palpebral ptosis and external ophthalmoplegia are rare in all groups of glycolytic enzyme-deficient CMS.

Deficiency of enzymes in O-mannosylation is observed in congenital muscular dystrophy including Fukuyama-type muscular dystrophy and is called dystroglycanopathy. Pathogenic variants of *GMPPB* cause muscular dystrophy-dystroglycanopathy (MDDG) type 14 [28] with hypoglycosylation of ɑ-dystroglycan and muscular dystrophy in biopsied muscle [25]. Muscle MRI shows displacement of muscle tissue to fibrous and adipose tissues in paravertebral and proximal skeletal muscles [25,407], as observed in muscular dystrophies. In *GMPPB*-CMS, serum CK is elevated to 2 to 24 times the upper limit of normal (average 10.7 times) [7,8]. In *GFPT1*-CMS, serum CK is elevated to about 3 times the upper limit of normal [7,8]. The same variant causes *GMPPB*-CMS [25] and limb-girdle muscle weakness [408], indicating that myasthenia might not be deeply evaluated.

ChEIs are usually effective for all groups of glycosylation-deficient CMS (*GFPT1-*CMS [390], *DPAGT1*-CMS [21], *ALG2*-CMS [24,401], *ALG14*-CMS [24,402], *GMPPB*-CMS [7,25,399]). However, ChEI had no effect on a single patient with *ALG2*-CMS [409]. Amifampridine was effective in *DPAGT1*-CMS [21,22]. Salbutamol (albuterol) was effective for *GFPT1*-CMS [410], *DPAGT1*-CMS [22], and *GMPPB*-CMS [25]. Ephedrine was effective for *ALG2*-CMS [409].

### 4.11. CMS Caused by Defective Nerve Terminal Formation (MYO9A and SLC25A1)

#### 4.11.1. Pathomechanisms

Myosin 9A (*MYO9A*) expressed in peripheral nerves is an atypical myosin carrying the Rho GTPase-activating protein (GAP) domain and regulates intracellular transport. Myo9a inhibits RHOA by stimulating its GTPase activity through the GAP domain [411]. Biallelic loss-of-function variants of *MYO9A* cause CMS [412]. Knockdown of two orthologs, *myo9aa/ab*, in zebrafish causes shortening and abnormal branching of spinal motor neurons, and defective NMJ signal transmission [412]. Agrin fragment rescued defective neurite elongation and motor deficits in *myo9aa/ab*-deficient zebrafish [413]. Knockdown of *Myo9a* in NSC34 cells revealed that Myo9a is essential for the formation and maintenance of neuronal cells, and for the transport of synaptic vesicles and protein secretion [413]. Interestingly, biallelic loss-of-function variants of *MYO9A* cause AMC [38]. In addition, hemiallelic premature termination codon of *MYO9A* causes focal segmental glomerulosclerosis [414].

Pathogenic variants of a succinate transporter (*SLC25A1*) in mitochondrial inner membrane cause combined D-2- and L-2-hydroxyglutaric aciduria (D2L2AD) [415] and CMS. Pathogenic variants of *SLC25A1* are predicted to compromise metabolisms of lipid, sterol synthesis, gluconeogenesis, and glycolysis [416], which somehow leads to the development of CMS. Knockdown of *Slc25a1* in zebrafish causes aberration in the axonal elongation of spinal motor neurons and compromise the formation of the NMJ [417]. *SLC25A1*-CMS is thus predicted to be caused by presynaptic defects.

#### 4.11.2. Clinical Features and Therapies

*MYO9A*-CMS was reported in 3 patients in 2 pedigrees in 2016 [412]. Patients were initially noted with reduced fetal movement before birth and palpebral ptosis at birth. Patients later developed swallowing difficulty, distal and proximal muscle weakness, episodic apnea, respiratory insufficiency, and external ophthalmoplegia. In two patients in a single pedigree, the presence of nystagmus was documented. All patients had developmental delay. ChEI was effective, and combination of ChEI and amifampridine showed marked effects in a patient. However, in a single patient, combination of amifampridine and fluoxetine induced respiratory crisis.

*SLC25A1*-CMS has been reported in 19 patients in 10 pedigrees since 2014 [417,418,419,420,421]. Limb myasthenia and palpebral ptosis are shared features. External ocular muscles, bulbar muscles, and respiratory muscles are sometimes affected. Developmental delays were also sometimes noted. ChEIs and amifampridine are generally ineffective but are slight effective in some patients.

### 4.12. CMS Caused by Defective Nuclear Membrane Protein (TOR1AIP1)

#### 4.12.1. Pathomechanisms

LAP1 is a ubiquitously expressed inner nuclear membrane protein. Its N-terminal domain interacts with A-type lamins and emerin in the nucleoplasm [422]. Its C-terminal luminal domain interacts with and activates nucleoplasmic TorsinA, an ATPase for the ATPases associated with diverse cellular activities (AAA+) [423]. Knockout of *Tor1aip1* in mouse shows endplate AChR deficiency with markedly increased number of myonuclei at the NMJ. Loss-of-function variants of *TOR1AIP1* were previously reported to cause limb-girdle muscular dystrophy or dystonia, with cardiomyopathy or a severe multisystem disorder [424,425,426]. Thus, CMS is a novel phenotype caused by pathogenic variants of *TOR1AIP1*. It is interesting to note that pathogenic variants of *LMNA* encoding another nucleolar membrane protein, lamin A, also cause multiple disease phenotypes.

#### 4.12.2. Clinical Features and Therapies

*TOR1AIP1*-CMS was reported in two adult siblings in 2020 [427] and three adult siblings in 2022 [428]. All patients were noted with mild to moderate muscle weakness and myasthenia in limb muscles and took a slowly progressive or stable course. ChEIs were effective [427,428], and addition of salbutamol (albuterol) had no effect [427].

### 4.13. CMS Caused by Defective Chromatin Remodeling Protein (CHD8)

#### 4.13.1. Pathomechanisms

CHD8 is one of ATP-dependent chromatin-remodeling enzymes but binds to β-catenin and suppresses the transcription of target genes of β-catenin [429,430,431]. CHD8 is accumulated at the NMJ and binds to rapsyn through β-catenin [432]. Thus, either transcriptional suppression of β-catenin-target genes or suppressed interaction between β-catenin and rapsyn is likely to account for *CHD8*-CMS [432]. In addition, knockout of *Ctnnb*1 encoding β-catenin (βCAT) that binds to CHD8 impairs AChR clustering and release of ACh from the nerve terminal [433]. In *Drosophila*, *Kis*, a homolog of *CHD8*, promoted presynaptic endocytosis at the NMJ [434]. Similarly, in *C. elegans*, a loss-of-function of *Chd8* caused reduced synaptic vesicle recycling [435]. As sated below, a marked effect of amifampridine and lack of effects of ChEIs and salbutamol (albuterol) are also consistent with the notion that the major defect in *CHD8*-CMS is at the motor nerve terminal [432].

#### 4.13.2. Clinical Features and Therapies

Monozygotic female twins with *CHD8*-CMS were reported in 2020 [432]. Patients showed neonatal onset respiratory distress, palpebral ptosis, and limb muscle weakness. At age 14 years, when the patients were reported, they showed frequent falling attacks and myasthenia, as well as rapidly progressive scoliosis. ChEI and salbutamol (albuterol) showed no effect, but amifampridine was markedly effective [432]. Hemiallelic loss-of-function variants of *CHD8* are also reported in intellectual developmental disorder with autism and macrocephaly (IDDAM) [436,437]. The authors of *CHD8*-CMS stated as personal communications that muscle hypotonia and muscle weakness were observed in 4 out of 66 patients with pathogenic variants of *CHD8* in IDDAM [432].

### 4.14. CMS in PURA Syndrome (PURA)

#### 4.14.1. Pathomechanisms

Purine-rich element-binding protein A (PURA, *PURA*) is involved in DNA replication, transcription, RNA transport, and mRNA translation, and is conserved across species. PURA plays essential roles in brain development, synapse formation, and proliferation of neuronal and glial cells. Hemiallelic loss-of-function variants of PURA were identified in 11 out of 2117 patients with neurodevelopmental delay in 2014 [438] and thereafter [439,440,441]. Analysis of 32 patients in the authors’ cohort and review of 22 reported patients with PURA syndrome showed that all patients had moderate to severe intellectual disability and neonate-onset symptoms including hypotonia (96%), respiratory problems (57%), feeding difficulties (77%), exaggerated startle response (44%), hypersomnolence (66%), hypothermia (35%), epilepsy (54%), gastrointestinal problems (69%), ophthalmological problems (51%), and endocrine problems (42%) [441]. PURA is expressed in many tissues and has many roles. The exact defects at the NMJ remain undetermined.

#### 4.14.2. Clinical Features and Therapies

Three patients with *PURA*-CMS showing fluctuating muscle weakness were reported in 2022 [55,442]. One patient showed decremental CAMP, as well as R-CMAP that was much higher than that observed in *COLQ*-CMS and SCCMS [55]. Another patient showed decremental CAMP followed by incremental CMAP [55]. The third patient showed no decremental CMAP at 3 Hz nerve stimulation but showed non-significant incremental CMAP at 30 Hz stimulation [442]. Two patients were neonates [55,442] and the other was 5 years old [55]. The 5-year-old patient became free of symptoms indicating defective NMJ signal transmission after age 2 years. In a patient, ChEI was ineffective, but salbutamol (albuterol) was effective and unnecessitated non-invasive positive pressure ventilation (NIPPV) because of amelioration of episodic apnea [55]. In another patient, ChEI had markedly ameliorated motor deficits [442].

## 5. Conclusions

CMS is a group of heterogenous disorders with highly variable clinical phenotypes that require specific treatment for specific pathomechanisms (Table 1). A total of 35 genes have been identified to cause CMS. Clinically overt major phenotypes of recently identified *TOR1AIP1*-CMS, *CHD8*-CMS, and *PURA*-CMS, as well as *GMPPB*-CMS reported in 2015 and *PLEC*-CMS in 1989 are not pure CMS. Indeed, pathogenic variants of these genes were initially reported to cause other diseases. Although not all these patients show defective NMJ signal transmission, the presence of defective NMJ was noted by detailed clinical and electrophysiological examinations. Scrutinizing analysis of the NMJ in other diseases may disclose additional groups of CMS in the future.

## Figures and Tables

**Figure 1 ijms-24-03730-f001:**
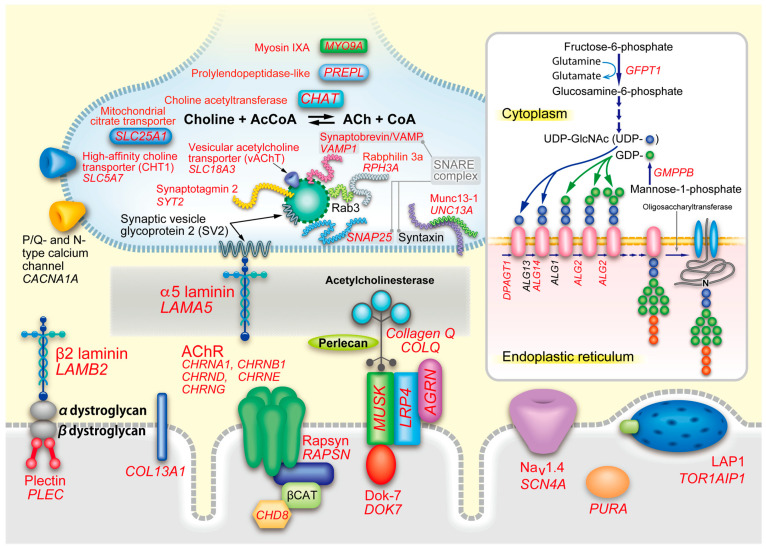
Thirty-five genes (red letters) causing CMS.

**Figure 2 ijms-24-03730-f002:**
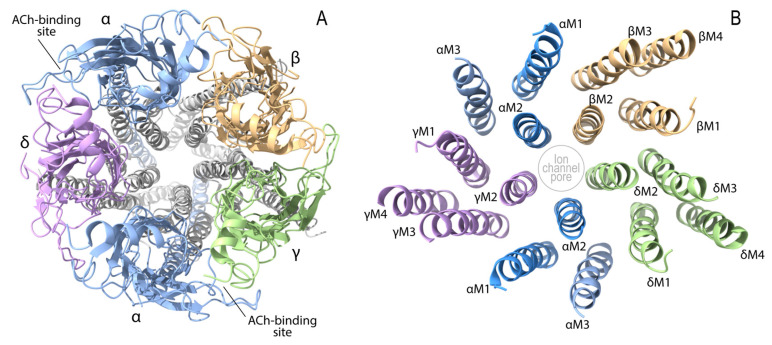
Crystal structure of AChR viewed from the extracellular side (PDB 2BG9) [87]. (**A**) Extracellular domains of AChR subunits. Other domains are shown in gray. (**B**) Transmembrane domains of AChR subunits. αM4 domains are not indicated.

**Figure 3 ijms-24-03730-f003:**
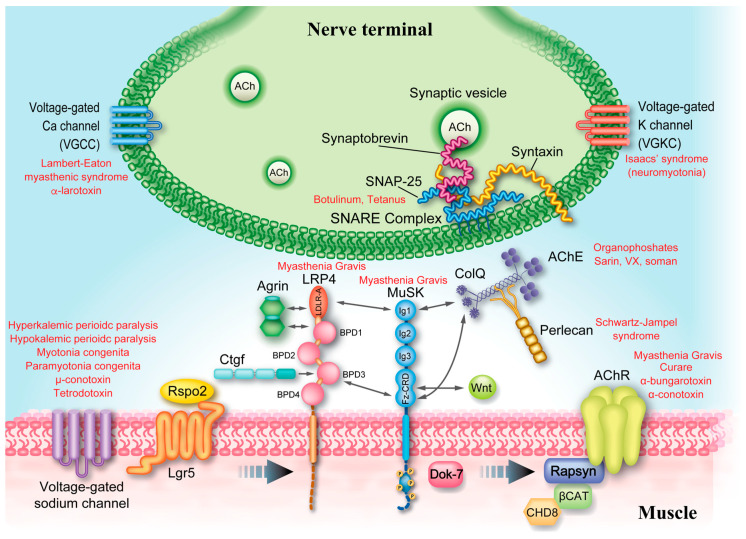
Representative molecules at the nerve terminal and the agrin-LPR4-MuSK signaling pathway to induce AChR clustering. Interactions between binding domains are indicated by double headed arrows [71,98,100,101,102]. Broken arrows in the muscle indicate that the exact signaling molecules are not shown. Diseases other than CMS and toxins affecting the NMJ are indicated in red letters. βCAT, β-catenin; BPD, β-propeller domain; C6, six-cysteine-box; Ctgf, connective tissue growth factor; Fz-CRD, frizzled-like cysteine-rich domain; Ig, immunoglobulin-like domain; LDLR-A, low-density lipoprotein receptor class A repeat; Lgr5, leucine-rich repeat-containing G-protein coupled receptor 5; and Rspo2, R-spondin 2.

**Figure 4 ijms-24-03730-f004:**
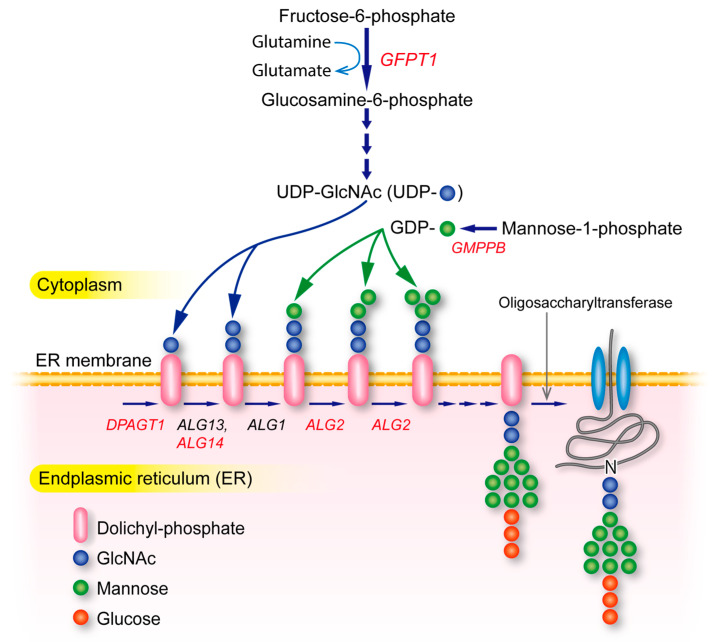
Hexosamine biosynthesis pathway to make UDP-GlcNAc and N-glycosylation pathway. Genes causing CMS are shown in red. Figure 4 is shown as an inset in Figure 1.

**Table 1 ijms-24-03730-t001:** Electrophysiological features and therapies of congenital myasthenic syndromes.

Section	Phenotype	Gene	OMIM		# ^a^	Inheritance	Low-Frequency RNS	High-Frequency RNS	Treatment						
									ChEIs	Ephedrine	Salbutamol (albuterol)	Amifampridine	Quinidine	Fluoxetine	Acetazolamide
4.1	Endplate AChR deficiency	*CHRNA1*			- ^b^	AR	decrement		effective	effective	effective	effective			
4.1	Endplate AChR deficiency	*CHRNB1*	CMS2C		- ^b^	AR	decrement		effective	effective	effective	effective			
4.1	Endplate AChR deficiency	*CHRND*	CMS3C		- ^b^	AR	decrement		effective	effective	effective	effective			
4.1	Endplate AChR deficiency	*CHRNE*	CMS4C		- ^b^	AR	decrement		effective	effective	effective	effective			
4.1	Endplate AChR deficiency	*RAPSN*	CMS11		[38]	AR	decrement		effective	effective	effective	effective			
4.2	Escobar syndrome	*CHRNG*			101	AR	decrement								
4.2	FADS	*CHRNA1*			(4)	AR	decrement								
4.2	FADS	*CHRND*			(6)	AR	decrement								
4.2	FADS	*MUSK*			(6)	AR	decrement								
4.2	FADS	*RAPSN*			(8)	AR	decrement								
4.2	FADS	*DOK7*			(2)	AR	decrement								
4.2	FADS	*SLC18A3*			(1)	AR	decrement								
4.3	SCCMS	*CHRNA1*	CMS1A		(14)	AD	decrement, repetitive CMAP		mostly ineffective	effective in some reports	effective in some reports		effective	effective	
4.3	SCCMS	*CHRNB1*	CMS2A		(5)	AD	decrement, repetitive CMAP		mostly ineffective	effective in some reports	effective in some reports		effective	effective	
4.3	SCCMS	*CHRND*	CMS3A		(4)	AD	decrement, repetitive CMAP		mostly ineffective	effective in some reports	effective in some reports		effective	effective	
4.3	SCCMS	*CHRNE*	CMS4A		(11)	AD/AR	decrement, repetitive CMAP		mostly ineffective	effective in some reports	effective in some reports		effective	effective	
4.3	FCCMS	*CHRNA1*	CMS1B		(3) ^c^	AR	decrement		effective	presumably effective, but no report	effective in a report	effective			
4.3	FCCMS	*CHRNB1*	CMS2B		(1) ^c^	AR	decrement		effective	presumably effective, but no report	effective in a report	effective			
4.3	FCCMS	*CHRND*	CMS3B		(1) ^c^	AR	decrement		effective	presumably effective, but no report	effective in a report	effective			
4.3	FCCMS	*CHRNE*	CMS4B		(6) ^c^	AR	decrement		effective	presumably effective, but no report	effective in a report	effective			
4.4	Endplate AChE deficiency	*COLQ*	CMS5		[30]	AR	decrement, repetitive CMAP		contraindication, but effective in some reported patients	effective in some reports	effective in some reports			effective in a report	
4.4	Synaptic CMS	*LAMB2*			1	AR	decrement		contraindication	effective					
4.4	Synaptic CMS	*COL13A1*	CMS19		41	AR	decrement		ineffective		effective	effective			
4.5	Sodium channel CMS	*SCN4A*	CMS16		6	AR	no decrement	decrement	Effective, ineffective, or marked adverse effects		Slightly effective				effective or ineffective
4.6	CMS caused by defective AChR clustering	*AGRN*	CMS8		[13]	AR	decrement		Ineffective or mildly effective	effective	effective	ineffective or slightly effective			
4.6	CMS caused by defective AChR clustering	*MUSK*	CMS9		[15]	AR	decrement		Ineffective or worsened		effective	effective			
4.6	CMS caused by defective AChR clustering	*LRP4*	CMS17		1	AR	decrement		worsened						
4.6	CMS caused by defective AChR clustering	*DOK7*	CMS10		[34]	AR	decrement		Combination of ineffective and worsening	effective	effective	effective in some reports		effective in some reports	
4.7	CMS caused by defective structural molecules	*PLEC*			22	AR	decrement		effective or ineffective		effective in some reports	effective or ineffective			
4.8	CMS caused by defective recycling of ACh	*CHAT*	CMS6		[19]	AR	no decrement	decrement in some patients	effective			effective			
4.8	CMS caused by defective recycling of ACh	*SLC18A3*	CMS21		7	AR	decrement at rest or only after isometric muscle contraction	decrement in some patients	effective	effective		effective			
4.8	CMS caused by defective recycling of ACh	*SLC5A7*	CMS20		12	AR	decrement at rest or only after isometric muscle contraction	decrement in some patients	effective	effective		ineffective			
4.8	CMS caused by defective recycling of ACh	*PREPL*	CMS22		18	AR	decrement	decrement in some patients	effective						
4.9	LEMS-like CMS	*SYT2*	CMS7ACMS7B		2	AD/AR	decrement	increment	effective		effective	effective			
4.9	LEMS-like CMS	*SNAP25*	CMS18		2	AD	decrement		ineffective		effective				
4.9	LEMS-like CMS	*UNC13A*			1	AR	decrement	increment	minimally effective			minimally effective			
4.9	LEMS-like CMS	*VAMP1*	CMS25		9	AR	decrement	increment	effective						
4.9	LEMS-like CMS	*RPH3A*			1	AR	no decrement	increment			effective				
4.9	LEMS-like CMS	*LAMA5*			1	AR	decrement	increment	effective			effective			
4.10	Glycosylation-deficient CMS	*GFPT1*		CMS12	[17]	AR	decrement		effective		effective				
4.10	Glycosylation-deficient CMS	*DPAGT1*		CMS13	[5]	AR	decrement		effective		effective	effective			
4.10	Glycosylation-deficient CMS	*ALG2*		CMS14	9	AR	decrement		effective or ineffective		effective	effective			
4.10	Glycosylation-deficient CMS	*ALG14*		CMS15	12	AR	decrement		effective						
4.10	Glycosylation-deficient CMS	*GMPPB*			[9]	AR	decrement		effective		effective				
4.11	CMS caused by defective nerve terminal formation	*MYO9A*		CMS24	3	AR	decrement		effective			effective			
4.11	CMS caused by defective nerve terminal formation	*SLC25A1*		CMS23	19	AR	decrement		ineffective in most patients			ineffective in most patients			
4.12	CMS caused by defective nuclear membrane protein	*TOR1AIP1*			5	AR	decrement		effective		no additional effect				
4.13	CMS caused by defective chromatin remodeling protein	*CHD8*			2	AR	decrement		ineffective		ineffective	markedly effective			
4.14	CMS in PURA syndrome	*PURA*			3	AD	decrement, repetitive CMAP		effective in a patient, but not in another patient		effective in a patient				

^a^ The number of original reports is shown in square brackets, and the number of pathogenic variants in round brackets. Otherwise, the number of patients is shown. ^b^ Differentiation of endplate AChR deficiency and FCCMS requires detailed electrophysiological studies using either intracellular recordings or patch-clamp recordings of biopsied patient’s neuromuscular junction, or patch-clamp recordings of mutant AChRs expressed in culture cells, but most variants are not characterized as such. Thus, the numbers of patients, original articles, pathogenic variants of endplate AChR deficiency were not counted. ^c^ For FCCMS, the number of pathogenic variants with electrophysiological analyses was counted.

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
