# Peer review of "Clinical and Pathologic Features of Congenital Myasthenic Syndromes Caused by 35 Genes—A Comprehensive Review"

_ijms, 2023, doi:10.3390/ijms24043730_

Round 1

Reviewer 1 Report

The manuscript is very well written and is an exemplary collection of all the information related to the clinical and pathological characteristics of the different forms of CMS. The figures of the NMJ are very nice and all the genes/proteins involved in CMS are very well described. 

Small clarifications to the authors:

Line 120: there are 2 full stops.

Line 211: the authors could insert the CACNA1A gene in figure 1

Line 212: the authors could insert the SNARE complex in figure 1

Line 450: remove in

Line 1016: the authors could put in brackets (bCAT) like in figure 3

Author Response

Comments by Reviewer #1

The manuscript is very well written and is an exemplary collection of all the information related to the clinical and pathological characteristics of the different forms of CMS. The figures of the NMJ are very nice and all the genes/proteins involved in CMS are very well described.

Small clarifications to the authors:

Line 120: there are 2 full stops.

Line 211: the authors could insert the CACNA1A gene in figure 1

Line 212: the authors could insert the SNARE complex in figure 1

Line 450: remove in

Line 1016: the authors could put in brackets (bCAT) like in figure 3

Answer to Reviewer #1

We appreciate productive and encouraging comments. We revised our manuscript to comply with the valuable suggestions. We also added CACNA1A and the SNARE complex in Fig. 1.

Reviewer 2 Report

The manuscript of Ohno et al. is a well-written, comprehensive summary of the CMS. The introduction provides an overview that is understandable for non-specialists. Figure 1 helps to understand the localization of the proteins involved in the CMS syndromes.

The next paragraphs are logically structured and give a necessary background, especially for people who are not well acquainted with CMS. The authors mention all relevant aspects, such as electrophysiology, CK, muscle biopsy, molecular background and therapy. Some information, such as the fact of missing ptosis or day-to-day muscle weakness fluctuation, are important for a diagnosis in the case of not-specialized centers. In the next part, the authors characterize the CMS related to the particular genetic variant. Conclusions are concise and summarize the content of the manuscript.

There are just a few small things that I would like to mention:

1. Line 166-168 Founder effect has also been reported in GMPPB (DOI: 10.1007/s10048-021-00658-1) or in the PLEC (doi: 10.3390/genes11070716) - the latter manuscript is already cited, but not in this content

2. Table 1. CMS in PURA syndrome-it is difficult to figure out in pdf format what treatment is effective and which is not (this will be corrected in the web version). There is one case report where treatment with pyridostigmine was successful DOI: 10.1016/j.nmd.2022.01.005 and two cases where pyridostigmine was not successful. In one of these cases, salbutamol showed a good effect DOI: 10.1016/j.nmd.2022.09.007

3. There is one aspect of CMS that I always found interesting, but it is not mentioned here (however, it is facultative). It seems that in a very few cases symptomatic heterozygosity may take place, sometimes only with congenital ptosis, sometimes with more pronounced symptoms. e.g.  

https://doi.org/10.1016/j.nmd.2006.11.010, https://doi.org/10.1093/brain/awm068, https://doi.org/10.1016/j.nmd.2020.02.009

Unfortunately, most of such cases are not reported.

Author Response

We appreciate productive and encouraging comments by reviewers. We revised our manuscript to comply with the valuable suggestions.

Comments by Reviewer #2

The manuscript of Ohno et al. is a well-written, comprehensive summary of the CMS. The introduction provides an overview that is understandable for non-specialists. Figure 1 helps to understand the localization of the proteins involved in the CMS syndromes. The next paragraphs are logically structured and give a necessary background, especially for people who are not well acquainted with CMS. The authors mention all relevant aspects, such as electrophysiology, CK, muscle biopsy, molecular background and therapy. Some information, such as the fact of missing ptosis or day-to-day muscle weakness fluctuation, are important for a diagnosis in the case of not-specialized centers. In the next part, the authors characterize the CMS related to the particular genetic variant. Conclusions are concise and summarize the content of the manuscript. There are just a few small things that I would like to mention:

Comment #2-1

Line 166-168 Founder effect has also been reported in GMPPB (DOI: 10.1007/s10048-021-00658-1) or in the PLEC (doi: 10.3390/genes11070716) - the latter manuscript is already cited, but not in this content

Answer #2-1

Thank you for the information. We added relevant information by citing the two articles.

Comment #2-2

Table 1. CMS in PURA syndrome-it is difficult to figure out in pdf format what treatment is effective and which is not (this will be corrected in the web version). There is one case report where treatment with pyridostigmine was successful DOI: 10.1016/j.nmd.2022.01.005 and two cases where pyridostigmine was not successful. In one of these cases, salbutamol showed a good effect DOI: 10.1016/j.nmd.2022.09.007

Answer #2-2

We apologize for obscure information on PURA-CMS in Table 1. We indicated that ChEI was effective in a patient but not in another patient, and that salbutamol was effective in a patient. We also added a header at the top of each page of Table 1 to clarify which column represents which information.

Comment #2-3

There is one aspect of CMS that I always found interesting, but it is not mentioned here (however, it is facultative). It seems that in a very few cases symptomatic heterozygosity may take place, sometimes only with congenital ptosis, sometimes with more pronounced symptoms. e.g. 

https://doi.org/10.1016/j.nmd.2006.11.010, https://doi.org/10.1093/brain/awm068, https://doi.org/10.1016/j.nmd.2020.02.009

Unfortunately, most of such cases are not reported.

Answer #2-3

We appreciate valuable information. We addressed the presence of possible symptomatic heterozygous carriers for COLQ-CMS, and the presence of heterozygous DOK7-CMS patients. We agree that lack of the second mutation on another allele prevents us from reporting patients with a hemiallelic mutation. We also addressed the presence of possible publication bias.